# Bayesian Learning via $Q$-Exponential Process

Shuyi Li    Michael O'Connor    Shiwei Lan [*]

*School of Mathematical & Statistical Sciences*
*Arizona State University, Tempe, AZ 85287*

## Abstract

Regularization is one of the most fundamental topics in optimization, statistics and machine learning. To get sparsity in estimating a parameter $u \in \mathbb{R}^d$, an $\ell_q$ penalty term, $\|u\|_q$, is usually added to the objective function. What is the probabilistic distribution corresponding to such $\ell_q$ penalty? What is the *correct* stochastic process corresponding to $\|u\|_q$ when we model functions $u \in L^q$? This is important for statistically modeling high-dimensional objects such as images, with penalty to preserve certain properties, e.g. edges in the image. In this work, we generalize the $q$-exponential distribution (with density proportional to) $\exp\left(-\frac{1}{2}|u|^q\right)$ to a stochastic process named *q-exponential (Q-EP) process* that corresponds to the $L_q$ regularization of functions. The key step is to specify consistent multivariate $q$-exponential distributions by choosing from a large family of elliptic contour distributions. The work is closely related to Besov process which is usually defined in terms of series. Q-EP can be regarded as a definition of Besov process with explicit probabilistic formulation, direct control on the correlation strength, and tractable prediction formula. From the Bayesian perspective, Q-EP provides a flexible prior on functions with sharper penalty ($q < 2$) than the commonly used Gaussian process (GP, $q = 2$). We compare GP, Besov and Q-EP in modeling functional data, reconstructing images and solving inverse problems and demonstrate the advantage of our proposed methodology.

## 1   INTRODUCTION

Regularization on function spaces is one of the fundamental questions in statistics and machine learning. High-dimensional objects such as images can be viewed as discretized functions defined on 2d or 3d domains. Statistical models for these objects on function spaces demand regularization to induce sparsity, prevent over-fitting, produce meaningful reconstruction, etc. Gaussian process [GP 38, 24] has been widely used as an $L_2$ penalty (negative log-density as a quadratic form) or a prior on the function space. Despite the flexibility, sometimes random candidate functions drawn from GP are over-smooth for modeling certain objects such as images with sharp edges. To address this issue, researchers have proposed a class of $L_1$ penalty based priors including Laplace random field [37, 34, 28] and Besov process [31, 15, 25, 16]. They have been extensively applied in spatial modeling [37], signal processing [28], imaging analysis [44, 34] and inverse problems [31, 15]. Figure 1 demonstrates an application of nonparametric regression models on functions endowed with GP, Besov and our proposed $q$-exponential process (Q-EP) priors respectively to reconstruct a blurry image of a satellite. Q-EP model generates the best reconstruction, indicating its advantage over GP in modeling objects with abrupt changes or sharp contrast such as "edges" in image.

For these high-dimensional (refer to its discretization) inhomogeneous objects on $d^\star$ domains $D \subset \mathbb{R}^{d^\star}$, particularly 2d images with sharp edges ($d^\star = 2$), one can model them as a random function $u$ from a

---

[*]slan@asu.edu

37th Conference on Neural Information Processing Systems (NeurIPS 2023).

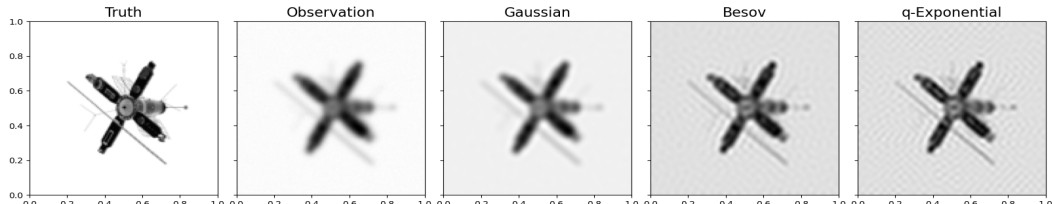

Figure 1: Image of satellite: true image, blurred observation, and reconstructions by GP, Besov and Q-EP models with relative errors 75.19%, 21.94% and 20.35% respectively.

Besov process represented by the following series for a given orthonormal basis $\{\phi_\ell\}_{\ell=1}^\infty$ in $L^2(D)$ [31, 15]:

$$u : D \longrightarrow \mathbb{R}, \quad u(x) = \sum_{\ell=1}^\infty \gamma_\ell u_\ell \phi_\ell(x), \quad u_\ell \overset{iid}{\sim} \pi_q(\cdot) \propto \exp\left(-\frac{1}{2}|\cdot|^q\right) \tag{1}$$

where $q \geq 1$ and $\gamma_\ell = \kappa^{-\frac{1}{q}} \ell^{-(\frac{s}{d^\star} + \frac{1}{2} - \frac{1}{q})}$ with (inverse) variance $\kappa > 0$ and smoothness $s > 0$. When $q = 2$ and $\{\phi_\ell\}$ is chosen to be Fourier basis, this reduces to GP [16] but Besov is often used with $q = 1$ and wavelet basis [31] to provide "edge-preserving" function candidates suitable for image analysis. Historically, [32] discovered that the total variation prior degenerates to GP prior as the discretization mesh becomes denser and thus loses the edge-preserving properties in high dimensional applications. Therefore, [31] proposed the Besov prior defined as in (1) and proved its discretization-invariant property. Though straightforward, such series definition lacks a direct way to specify the correlation structure as GP does through the covariance function. What is more, once the basis $\{\phi_\ell\}$ is chosen, there is no natural way to make prediction with Besov process.

We propose a novel stochastic process named *q-exponential process (Q-EP)* to address these issues. We start with the *q*-exponential distribution $\pi_q(\cdot)$ and generalize it to a multivariate distribution (from a family of elliptic contour distributions) that is consistent to marginalization. Such consistency requires the joint distribution and the marginalized one (by any subset of components) to have the same format of density (See Section 3). We then generalize such multivariate *q*-exponential distribution to the process Q-EP and establish its connection and contrast to the Besov process. Note, if we view the negative log-density of the proposed distribution and process, Q-EP would impose an $L_q$ regularization on the function space, similarly as $L_2$ regularization given by GP whose negative log-density is a quadratic form of the input variable $x$ (See Remark 2).

**Connection to existing works**  The proposed Q-EP process is related to the student-*t* process (TP) [41] as alternatives to GP. TP generalizes multivariate *t*-distribution (MVT) and is derived as a scale mixture of GP. Both TP and Q-EP can be viewed as a special case of the elliptical process [5] which gives the condition on general elliptic distributions that can be generalized to a valid stochastic process. Both papers focus on extending GP to robust models for heavier tail data, while our proposed work innovates a new Bayesian learning method on function spaces through the regularization parameter *q* (See Figure 3 for its effect on regularization when it varies), as is usually done in the optimization. Both our proposed Q-EP and [5] are inspired by Kano's consistency result [27], however the later focuses on a completely different process named squeezebox process. Our work on Q-EP makes multi-fold contributions to the learning of functional data in statistics and machine learning:

1. We propose a novel stochastic process Q-EP corresponding to the $L_q$ regularization on function spaces.
2. For the first time we define/derive Besov process probabilistically as Q-EP with direct ways to configure correlation and to make prediction.
3. We provide flexible Bayesian inference methods based on the Markov Chain Monte Carlo (MCMC) algorithms using a white-noise representation for Q-EP prior models.

The rest of the paper is organized as follows. Section 2 introduces the *q*-exponential distribution and its multivariate generalizations. We propose the Q-EP with details in Section 3 and introduce it as a nonparametric prior for modeling functional data. In Section 4 we demonstrate the advantage of Q-EP over GP and Besov in time series modeling, image reconstruction, and Bayesian inverse problems (Appendix C.4). Finally we discuss some future directions in Section 5.

## 2 THE *Q*-EXPONENTIAL DISTRIBUTION AND ITS MULTIVARIATE GENERALIZATIONS

Let us start with the *q*-exponential distribution for a scalar random variable $u \in \mathbb{R}$. It is named in [15] and defined with the following density not in an exact form (as a probability density normalized to 1):

$$\pi_q(u) \propto \exp\left(-\frac{1}{2}|u|^q\right). \tag{2}$$

This *q*-exponential distribution (2) is actually a special case of the following *exponential power (EP)* distribution $EP(\mu, \sigma, q)$ with $\mu = 0$, $\sigma = 1$:

$$p(u|\mu, \sigma, q) = \frac{q}{2^{1+1/q}\sigma\Gamma(1/q)}\exp\left\{-\frac{1}{2}\left|\frac{u-\mu}{\sigma}\right|^q\right\} \tag{3}$$

where $\Gamma$ denotes the gamma function. Note the parameter $q > 0$ in (3) controls the tail behavior of the distribution: the smaller $q$ the heavier tail and vice versa. This distribution also includes many commonly used ones such as the normal distribution $\mathcal{N}(\mu, \sigma^2)$ for $q = 2$ and the Laplace distribution $L(\mu, b)$ with $\sigma = 2^{-1/q}b$ when $q = 1$.

How can we generalize it to a multivariate distribution and further to a stochastic process? Gomez [23] provided one possibility of a multivariate EP distribution, denoted as $EP_d(\boldsymbol{\mu}, \mathbf{C}, q)$, with the following density:

$$p(\mathbf{u}|\boldsymbol{\mu}, \mathbf{C}, q) = \frac{q\Gamma(\frac{d}{2})}{2\Gamma(\frac{d}{q})}2^{-\frac{d}{q}}\pi^{-\frac{d}{2}}|\mathbf{C}|^{-\frac{1}{2}}\exp\left\{-\frac{1}{2}\left[(\mathbf{u}-\boldsymbol{\mu})^{\mathsf{T}}\mathbf{C}^{-1}(\mathbf{u}-\boldsymbol{\mu})\right]^{\frac{q}{2}}\right\} \tag{4}$$

When $q = 2$, it reduces to the familiar multivariate normal (MVN) distribution $\mathcal{N}_d(\boldsymbol{\mu}, \mathbf{C})$.

Unfortunately, unlike MVN being the foundation of GP, the Gomez's EP distribution $EP_d(\boldsymbol{\mu}, \mathbf{C}, q)$ fails to generalize to a valid stochastic process because it does not satisfy the marginalization consistency as MVN does (See Section 3 for more details). It turns out we need to seek candidates in an even larger family of *elliptic* (contour) distributions $EC_d(\boldsymbol{\mu}, \mathbf{C}, g)$:

**Definition 2.1** (Elliptic distribution). *A multivariate elliptic distribution $EC_d(\boldsymbol{\mu}, \mathbf{C}, g)$ has the following density [26]*

$$p(\mathbf{u}) = k_d|\mathbf{C}|^{-\frac{1}{2}}g(r), \quad r(\mathbf{u}) = (\mathbf{u}-\boldsymbol{\mu})^{\mathsf{T}}\mathbf{C}^{-1}(\mathbf{u}-\boldsymbol{\mu}) \tag{5}$$

*where $k_d > 0$ is the normalizing constant and $g(\cdot)$, a one-dimensional real-valued function independent of $d$ and $k_d$, is named* density generating function *[19]*.

Every elliptic (contour) distributed random vector $\mathbf{u} \sim EC_d(\boldsymbol{\mu}, \mathbf{C}, g)$ has a stochastic representation mainly due to Schoenberg [40, 12, 26], as stated in the following theorem.

**Theorem 2.1.** $\mathbf{u} \sim EC_d(\boldsymbol{\mu}, \mathbf{C}, g)$ *if and only if*

$$\mathbf{u} \overset{d}{=} \boldsymbol{\mu} + R\mathbf{L}S \tag{6}$$

*where $S \sim \mathrm{Unif}(\mathscr{S}^{d+1})$ uniformly distributed on the unit-sphere $\mathscr{S}^{d+1}$, $\mathbf{L}$ is the Cholesky factor of $\mathbf{C}$ such that $\mathbf{C} = \mathbf{L}\mathbf{L}^{\mathsf{T}}$, $R \perp S$ and $R^2 \overset{d}{=} r(\mathbf{u}) \sim f(r) = \frac{\pi^{\frac{d}{2}}}{\Gamma(\frac{d}{2})}k_d r^{\frac{d}{2}-1}g(r)$.*

The Gomez's EP distribution $EP_d(\boldsymbol{\mu}, \mathbf{C}, q)$ is a special elliptic distribution $EC_d(\boldsymbol{\mu}, \mathbf{C}, g)$ with $g(r) = \exp\{-\frac{1}{2}r^{\frac{q}{2}}\}$ and $R^q \sim \Gamma(\alpha = \frac{d}{q}, \beta = \frac{1}{2})$ [23]. Not all elliptical distributions can be used to create a valid process [5]. In the following, we will carefully choose the density generator $g$ in $EC_d(\boldsymbol{\mu}, \mathbf{C}, g)$ to define a consistent multivariate *q*-exponential distribution generalizable to a process appropriately.

## 3 THE *Q*-EXPONENTIAL PROCESS

To generalize $EC_d(\boldsymbol{\mu}, \mathbf{C}, g)$ to a valid stochastic process, we need to choose proper $g$ such that the resulting distribution satisfies two conditions of Kolmogorov extension theorem [35]:

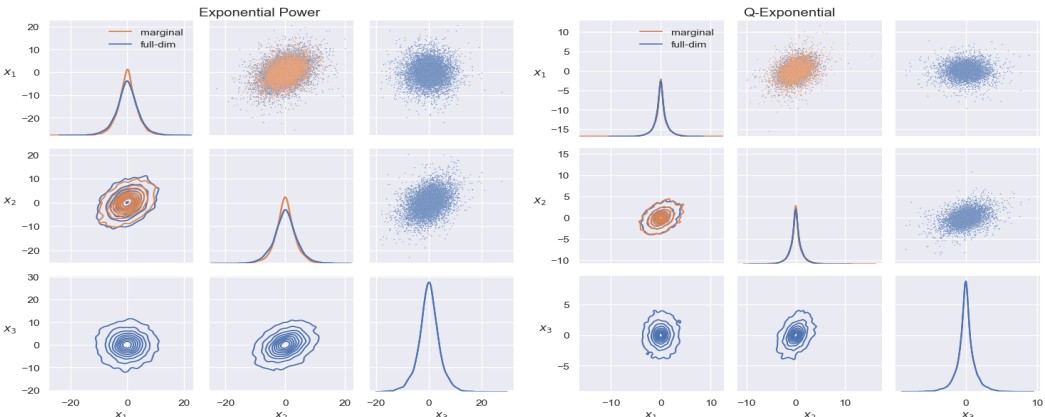

Figure 2: Inconsistent (Gomez's) EP distribution $EP_d(\boldsymbol{\mu}, \mathbf{C}, q)$ (left) vs. consistent Q-exponential distribution $q-ED_d(\boldsymbol{\mu}, \mathbf{C})$ (right). Both can be sampled using (6) with $R^q \sim \Gamma(\alpha = \frac{d}{q}, \beta = \frac{1}{2})$ and $R^q \sim \Gamma(\alpha = \frac{d}{2}, \beta = \frac{1}{2})$ respectively. Note there is significant discrepancy between the marginalization of $EP_3(\boldsymbol{\mu}, \mathbf{C}, q)$ and $EP_2(\boldsymbol{\mu}, \mathbf{C}, q)$. However, the marginalization of $q-ED_3(\boldsymbol{\mu}, \mathbf{C})$ coincides with $q-ED_2(\boldsymbol{\mu}, \mathbf{C})$. Empirical densities are estimated based on 10000 samples (shown as dots) for $q = 1$.

**Theorem 3.1** (Kolmogorov's Extension). *For all $t_1, \cdots, t_k \in T$, $k \in \mathbb{N}$ let $\nu_{t_1, \cdots, t_k}$ be probability measures on $\mathbb{R}^{dk}$ satisfying*

$$(K1): \nu_{t_{\sigma(1)}, \cdots, t_{\sigma(k)}}(F_1 \times \cdots \times F_k) = \nu_{t_1, \cdots, t_k}(F_{\sigma^{-1}(1)} \times \cdots \times F_{\sigma^{-1}(k)}) \, for \, all \, permutations \, \sigma \in S(k)$$

$$(K2): \nu_{t_1, \cdots, t_k}(F_1 \times \cdots \times F_k) = \nu_{t_1, \cdots, t_k, t_{k+1}, \cdots, t_{k+m}}(F_1 \times \cdots \times F_k \times \mathbb{R}^d \times \cdots \times \mathbb{R}^d) \, for \, all \, m \in \mathbb{N}$$

$$(7)$$

*Then there exists a probability space $(\Omega, \mathscr{F}, P)$ and a stochastic process $\{X_t\}$ on $\Omega$, $X_t : \Omega \to \mathbb{R}^n$ such that*

$$\nu_{t_1, \cdots, t_k}(F_1 \times \cdots \times F_k) = P[X_{t_1} \in F_1, \cdots, X_{t_k} \in F_k]$$

$$(8)$$

*for all $t_i \in T$, $k \in \mathbb{N}$ and all Borel sets $F_i \in \mathscr{F}$. (K1) and (K2) are referred to as **exchangeability** and **consistency** conditions respectively.*

As pointed out by Kano [27], the elliptic distribution $EC_d(\boldsymbol{\mu}, \mathbf{C}, g)$ in the format of Gomez's EP distribution (4) with $g(r) = \exp\{-\frac{1}{2}r^{\frac{q}{2}}\}$ does not satisfy the consistency condition [also c.f. Proposition 5.1 of 23]. Figure 2 (left panel) also illustrates such inconsistency numerically. However, Kano's consistency theorem [27] suggests a different viable choice of $g$ to make a valid generalization of $EC_d(\boldsymbol{\mu}, \mathbf{C}, g)$ to a stochastic process [5]:

**Theorem 3.2** (Kano's Consistency). *An elliptic distribution is* consistent *if and only if its density generator function, $g(\cdot)$, has the following form*

$$g(r) = \int_0^\infty \left(\frac{s}{2\pi}\right)^{\frac{d}{2}} \exp\left\{-\frac{rs}{2}\right\} p(s) ds$$

$$(9)$$

*where $p(s)$ is a strictly positive mixing distribution independent of $d$ and $p(s = 0) = 0$.*

### 3.1 Consistent Multivariate $Q$-exponential Distribution

In the above theorem 3.2, if we choose $p(s) = \delta_{r^{\frac{q}{2}-1}}(s)$, then we have $g(r) = r^{(\frac{q}{2}-1)\frac{d}{2}} \exp\left\{-\frac{r^{\frac{q}{2}}}{2}\right\}$, which leads to the following consistent *multivariate q-exponential distribution* $q-ED_d(\boldsymbol{\mu}, \mathbf{C})$.

**Definition 3.1.** *A multivariate q-exponential distribution, denoted as $q-ED_d(\boldsymbol{\mu}, \mathbf{C})$, has the following density*

$$p(\mathbf{u}|\boldsymbol{\mu}, \mathbf{C}, q) = \frac{q}{2}(2\pi)^{-\frac{d}{2}}|\mathbf{C}|^{-\frac{1}{2}}\boxed{r^{(\frac{q}{2}-1)\frac{d}{2}}}\exp\left\{-\frac{r^{\frac{q}{2}}}{2}\right\}, \quad r(\mathbf{u}) = (\mathbf{u} - \boldsymbol{\mu})^\mathsf{T}\mathbf{C}^{-1}(\mathbf{u} - \boldsymbol{\mu}) \quad (10)$$

**Remark 1.** *When $q = 2$, q$-$ED$_d(\boldsymbol{\mu}, \mathbf{C})$ reduces to MVN $\mathcal{N}_d(\boldsymbol{\mu}, \mathbf{C})$. When $d = 1$, if we let $C = 1$, then we have the density for $u$ as $p(u) \propto |u|^{\frac{q}{2}-1} \exp\left\{-\frac{1}{2}|u|^q\right\}$, differing from the original un-normalized density $\pi_q$ in (2) by a term $|u|^{\frac{q}{2}-1}$. This is needed for the consistency of process generalization. Numerically, it has the similar "edge-preserving" property as the Besov prior.*

**Remark 2.** *If taken negative logarithm, the density of q$-$ED$_d$ in (10) yields a quantity dominated by some weighted $L_q$ norm of $\mathbf{u} - \boldsymbol{\mu}$, i.e. $\frac{1}{2}r^{\frac{q}{2}} = \frac{1}{2}\|\mathbf{u} - \boldsymbol{\mu}\|_{\mathbf{C}}^q$. From the optimization perspective, q$-$ED$_d$, when used as a prior, imposes $L_q$ regularization in obtaining the maximum a posterior (MAP).*

Regardless of the normalizing constant, our proposed multivariate $q$-exponential distribution q$-$ED$_d(\boldsymbol{\mu}, \mathbf{C})$ differs from the Gomez's EP distribution EP$_d(\boldsymbol{\mu}, \mathbf{C}, q)$ by a boxed term $\boxed{r^{(\frac{q}{2}-1)\frac{d}{2}}}$. As stated in the following theorem, q$-$ED$_d$ satisfies the two conditions of Kolmogorov extension theorem thus is ready to generalize to a stochastic process (See the right panel of Figure 2 for the consistency).

**Theorem 3.3.** *The multivariate q-exponential distribution is both **exchangeable** and **consistent**.*

*Proof.* See Appendix A.1. $\qquad\qquad\square$

Like student-*t* distribution [41] and other elliptic distributions [5], we can show (See Appendix A.5) that q$-$ED$_d$ is represented as a scale mixture of Gaussian distributions for $0 < q < 2$ [27, 3, 47].

Numerically, thanks to our choice of density generator $g(r) = r^{(\frac{q}{2}-1)\frac{d}{2}} \exp\left\{-\frac{r^{\frac{q}{2}}}{2}\right\}$, one can show that $R^q \sim \chi_d^2$ (as in Appendix A.4) thus $R$ in Theorem 2.1 can be sampled as $q$-root of a $\chi_d^2$ random variable, which completes the recipe for generating random vector $\mathbf{u} \sim$ q$-$ED$_d(0, \mathbf{C})$ based on the stochastic representation (6). This is important for the Bayesian inference as detailed in Section 3.3.1. Note the matrix $\mathbf{C}$ in the definition (10) characterizes the covariance between the components, as shown in the following proposition.

**Proposition 3.1.** *If $\mathbf{u} \sim$ q$-$ED$_d(\boldsymbol{\mu}, \mathbf{C})$, then we have*

$$\mathrm{E}[\mathbf{u}] = \boldsymbol{\mu}, \; \mathrm{Cov}(\mathbf{u}) = \frac{2^{\frac{2}{q}}\Gamma(\frac{d}{2} + \frac{2}{q})}{d\Gamma(\frac{d}{2})} \mathbf{C} \overset{\cdot}{\sim} d^{\frac{2}{q}-1}\mathbf{C}, \; as \; d \to \infty \tag{11}$$

*Proof.* See Appendix A.4. $\qquad\qquad\square$

## 3.2 *Q*-exponential Process as Probabilistic Definition of Besov Process

To generalize $\mathbf{u} \sim$ q$-$ED$_d(0, \mathbf{C})$ to a stochastic process, we need to scale it to $\mathbf{u}^* = d^{\frac{1}{2}-\frac{1}{q}}\mathbf{u}$ so that its covariance is asymptotically finite. If $\mathbf{u} \sim$ q$-$ED$_d(0, \mathbf{C})$, then we denote $\mathbf{u}^* \sim$ q$-$ED$_d^*(0, \mathbf{C})$ following a *scaled q*-exponential distribution. Let $\mathscr{C} : L^q \to L^q$ be a kernel operator in the trace class, i.e. having eigen-pairs $\{\lambda_\ell, \phi_\ell(x)\}_{\ell=1}^\infty$ such that $\mathscr{C}\phi_\ell(x) = \phi_\ell(x)\lambda_\ell$, $\|\phi_\ell\|_2 = 1$ for all $\ell \in \mathbb{N}$ and $\mathrm{tr}(\mathscr{C}) = \sum_{\ell=1}^\infty \lambda_\ell < \infty$. Now we are ready to define the *q-exponential process (Q-EP)* with the scaled $q$-exponential distribution.

**Definition 3.2** (Q-EP). *A (centered) q-exponential process $u(x)$ with kernel $\mathscr{C}$ in the trace class, q$-\mathscr{E}\mathscr{P}(0, \mathscr{C})$, is a collection of random variables such that any finite set, $\mathbf{u} = (u(x_1), \cdots u(x_d))$, follows a scaled multivariate q-exponential distribution, i.e. $\mathbf{u} \sim$ q$-$ED$_d^*(0, \mathbf{C})$.*

Note, the process is defined on the $d^\star$-dimensional space depending on the applications ($x \in \mathbb{R}^{d^\star}$); while $d$ refers to the dimension of discretized process ($\mathbf{u} \in \mathbb{R}^d$). While Q-EP reduces to GP when $q = 2$, $q = 1$ is often adopted for imaging analysis as an "edge-preserving" prior. Illustrated in Figure 3 for selected $q \in (0, 2]$, smaller $q$ leads to sharper image reconstruction with varying $q$ interpolating between different regularization effects. Both Besov and Q-EP are valid stochastic processes stemming from the $q$-exponential distribution $\pi_q$. They are both designed to generalize GP to have sharper regularization (through $q$) but Q-EP has advantages in 1) the capability of specifying correlation structure directly and 2) the tractable prediction formula.

It follows from (1) immediately that the covariance of the Besov process $u(\cdot)$ at two points $x, x' \in \mathbb{R}^{d^\star}$:

$$\mathrm{Cov}(u(x), u(x')) = \sum_{\ell=1}^\infty \gamma_\ell^2 \phi_\ell(x) \otimes \phi_\ell(x') \tag{12}$$

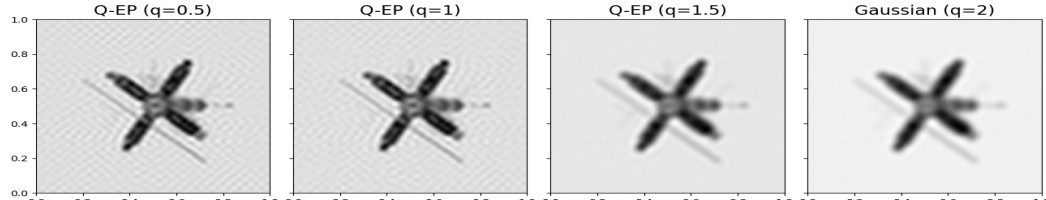

Figure 3: Image of satellite: MAP estimates by Q-EP with varying $q$ parameters.

Although the smoothness and correlation strength of Besov process can be configured by proper orthonormal basis $\{\phi_\ell\}$ as in (12), it is less straightforward than the kernel function working for GP. On the other hand, Q-EP has more freedom on the correlation structure through (11) with flexible choices from a large class of kernels including powered exponential, Matérn and periodic where we can directly specify the correlation length.

While Q-EP can be viewed as a probabilistic definition of Besov, the following theorem further establishes their connection in sharing equivalent series representations.

**Theorem 3.4** (Karhunen-Loéve). *If $u(x) \sim q-\mathscr{E}\mathscr{P}(0,\mathscr{C})$ with a trace class operator $\mathscr{C}$ having eigen-pairs $\{\lambda_\ell, \phi_\ell(x)\}_{\ell=1}^\infty$ such that $\mathscr{C}\phi_\ell(x) = \phi_\ell(x)\lambda_\ell$, $\|\phi_\ell\|_2 = 1$ for all $\ell \in \mathbb{N}$ and $\sum_{\ell=1}^\infty \lambda_\ell < \infty$, then we have the following series representation for $u(x)$:*

$$u(x) = \sum_{\ell=1}^\infty u_\ell \phi_\ell(x), \quad u_\ell := \int_D u(x)\phi_\ell(x) \overset{ind}{\sim} q-\mathrm{ED}^*(0,\lambda_\ell) \tag{13}$$

*where $\mathrm{E}[u_\ell] = 0$ and $\mathrm{Cov}(u_\ell, u_{\ell'}) = \lambda_\ell \delta_{\ell\ell'}$ with Dirac function $\delta_{\ell\ell'} = 1$ if $\ell = \ell'$ and $0$ otherwise.*

*Proof.* See Appendix A.5. $\qquad\square$

**Remark 3.** *If we factor $\sqrt{\lambda_\ell}$ out of $u_\ell$, we have the following expansion for Q-EP more comparable to (1) for Besov:*

$$u(x) = \sum_{\ell=1}^\infty \sqrt{\lambda_\ell} u_\ell \phi_\ell(x), \quad u_\ell \overset{iid}{\sim} q-\mathrm{ED}(0,1) \propto \pi_q(\cdot) \tag{14}$$

### 3.3 Bayesian Modeling with $Q$-exponential Process

Now let us consider the generic Bayesian regression model:

$$\begin{aligned} y &= u(x) + \varepsilon, \quad \varepsilon \sim L(\cdot; 0, \Sigma) \\ u &\sim \mu_0(du) \end{aligned} \tag{15}$$

where $L(\cdot; 0, \Sigma)$ denotes some likelihood model with zero mean and covariance $\Sigma$, and the mean function $u$ can be given a prior either Besov or Q-EP. Because of the definition (1) in terms of expanded series, there is no explicit formula for the posterior prediction using Besov prior. By contrast, a tractable formula exists for the posterior predictive distribution for (15) with Q-EP prior $\mu_0 = q-\mathscr{E}\mathscr{P}(0,\mathscr{C})$ when the likelihood happens to be $L(\cdot; 0, C) = q-\mathrm{ED}(\mathbf{0}, \mathbf{C})$, as stated in the following theorem.

**Theorem 3.5** (Posterior Prediction). *Given covariates $\mathbf{x} = \{x_i\}_{i=1}^N$ and observations $\mathbf{y} = \{y_i\}_{i=1}^N$ following $q-\mathrm{ED}$ in the model (15) with $q-\mathscr{E}\mathscr{P}$ prior for the same $q > 0$, we have the following posterior predictive distribution for $u(x_*)$ at (a) new point(s) $x_*$:*

$$u(x_*)|\mathbf{y}, \mathbf{x}, x_* \sim q-\mathrm{ED}(\boldsymbol{\mu}^*, \mathbf{C}^*), \quad \boldsymbol{\mu}^* = \mathbf{C}_*^\mathsf{T}(\mathbf{C}+\Sigma)^{-1}\mathbf{y}, \quad \mathbf{C}^* = \mathbf{C}_{**} - \mathbf{C}_*^\mathsf{T}(\mathbf{C}+\Sigma)^{-1}\mathbf{C}_* \tag{16}$$

*where $\mathbf{C} = \mathscr{C}(\mathbf{x},\mathbf{x})$, $\mathbf{C}_* = \mathscr{C}(\mathbf{x},x_*)$, and $\mathbf{C}_{**} = \mathscr{C}(x_*,x_*)$.*

*Proof.* See Appendix A.6. $\qquad\square$

**Remark 4.** *From Theorem 3.5, we know that Q-EP has same predictive mean as GP. But their predictive covariances differ by a constant $\left(\frac{2}{d}\right)^{\frac{2}{q}} \frac{\Gamma(\frac{d}{2}+\frac{2}{q})}{\Gamma(\frac{d}{2})}$ (asymptotically 1) based on Proposition 3.1.*

When the likelihood $L$ is not Q-EP, e.g. multinomial, such conjugacy is absent. Then we refer to the following sampling method for the posterior inference.

### 3.3.1 Inference by White-Noise MCMC

We follow [13] to consider the pushforward ($T$) of Gaussian white noise $\nu_0$ for non-Gaussian measures $\mu_0 = T^\# \nu_0$. More specifically, we construct a measurable transformation $T : \mathbf{z} \rightarrow \mathbf{u}$ that maps standard Gaussian random variables to $q$-exponential random variables. The transformation based on the stochastic representation (6) is more straightforward than that for Besov based on series expansions proposed by [13].

Recall the stochastic representation (6) of $\mathbf{u} \sim \mathrm{q-ED}_d(\mathbf{0}, \mathbf{C})$: $\mathbf{u} = R\mathbf{L}S$ with $R^q \sim \chi_d^2$ and $S \sim \mathrm{Unif}(\mathscr{S}^{d+1})$. We can rewrite $S \overset{d}{=} \frac{\mathbf{z}}{\|\mathbf{z}\|_2}$, $\quad R^q \overset{d}{=} \|\mathbf{z}\|_2^2$, $\quad$ for $\mathbf{z} \sim \mathscr{N}_d(\mathbf{0}, \mathbf{I}_d)$. Therefore, we have the pushforward mapping ($T$) and its inverse ($T^{-1}$) as

$$\mathbf{u} = T(\mathbf{z}) = \mathbf{L}\mathbf{z}\|\mathbf{z}\|^{\frac{2}{q}-1}, \quad \mathbf{z} = T^{-1}(\mathbf{u}) = \mathbf{L}^{-1}\mathbf{u}\|\mathbf{L}^{-1}\mathbf{u}\|^{\frac{q}{2}-1} \tag{17}$$

Figure B.1 illustrates that sampling with the white-noise representation (17) is indistinguishable from sampling by the stochastic representation (6). Then we can apply such white-noise representation to dimension-independent MCMC algorithms including preconditioned Crank-Nicolson (pCN) [14], infinite-dimensional Metropolis adjusted Langevin algorithm ($\infty$-MALA) [10], infinite-dimensional Hamiltonian Monte Carlo ($\infty$-HMC) [7], and infinite-dimensional manifold MALA ($\infty$-mMALA) [8] and HMC ($\infty$-mHMC) [9]. See Algorithm 1 for an implementation on pCN, hence named *white-noise pCN (wn-pCN)*.

### 3.3.2 Hyper-parameter Tuning

As in GP, there are hyper-parameters in the covariance function of Q-EP, e.g. variance magnitude ($\sigma^2$) and correlation length ($\rho$), that require careful adjustment and fine tuning. If the data process $y(x)$ and its mean $u(x)$ are both Q-EP with the same $q$, then we can have the marginal likelihood [38] as another Q-EP (c.f. Theorem 3.5). In general, when there is no such tractability, hyper-parameter tuning by optimizing the marginal likelihood is unavailable. However, we could impose conjugate hyper-priors on some parameters or even marginalize them to facilitate the inference of them (See Appendix A.7 for a proposition on such conditional conjugacy for the variance magnitude $\sigma^2$).

To tune the correlation length ($\rho$), we could impose a hyper-prior for $\rho$ and sample from $p(\rho|\mathbf{u})$. We then alternate updating $\mathbf{u}$ and hyper-parameters in a Gibbs scheme. In general, one could also use Bayesian optimization methods [33, 18, 4] for the hyper-parameter tuning.

## 4 NUMERICAL EXPERIMENTS

In this section, we compare GP, Besov and Q-EP by modeling time series (temporal), reconstructing images (spatial) from computed tomography and solving a (spatiotemporal) inverse problem (Appendix C.4). These numerical experiments demonstrate that our proposed Q-EP enables faster convergence in obtaining a better maximum a posterior (MAP) estimate. What is more, white-noise MCMC based inference provides appropriate uncertainty quantification (UQ) (by the posterior standard deviation). More numerical results can be found in the supplementary materials which also contain some demonstration codes. All the computer codes are publicly available at https://github.com/lanzithinking/Q-EXP.

### 4.1 Time Series Modeling

We first consider two simulated time series, one with step jumps and the other with sharp turnings, whose true trajectories are as follows:

$$u_J(t) = 1, \quad t \in [0,1]; \quad 0.5, \quad t \in (1,1.5]; \quad 2, \quad t \in (1.5,2]; \quad 0, \quad otherwise$$
$$u_T(t) = 1.5t, \quad t \in [0,1]; \quad 3.5 - 2t, \quad t \in (1,1.5]; \quad 3t - 4, \quad t \in (1.5,2]; \quad 0, \quad otherwise$$

We generate the time series $\{y_i\}$ by adding Gaussian noises to the true trajectories evaluated at $N = 200$ evenly spaced points $\{t_i\}$ in $[0,2]$, that is, $y_i^* = u_*(t_i) + \varepsilon_i$, $\varepsilon_i \overset{ind}{\sim} N(0, \sigma_*^2(t_i))$, $i = 1, \cdots, N$, $* = \mathrm{J}, \mathrm{T}$. Let $\sigma_J/\|u_J\| = 0.015$ *for* $t_i \in [0,2]$ and $\sigma_T/\|u_T\| = 0.01$ *if* $t_i \in [0,1]$; $0.07$ *if* $t_i \in (1,2]$. In addition, we also consider two real data sets of Tesla and Google stock prices in 2022. See Figures 4 (and Figures C.2) for the true trajectories (blue lines) and realizations (orange dots) respectively.

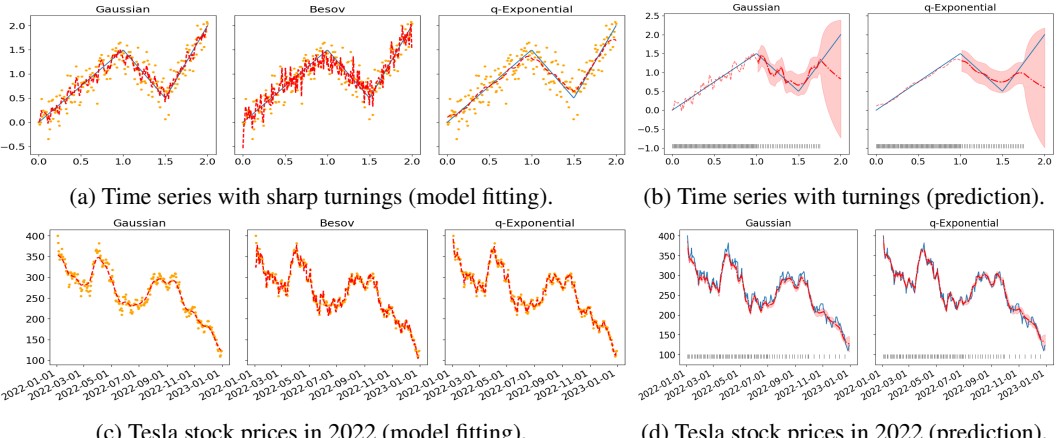

(a) Time series with sharp turnings (model fitting).

(b) Time series with turnings (prediction).

(c) Tesla stock prices in 2022 (model fitting).

(d) Tesla stock prices in 2022 (prediction).

Figure 4: (a)(c) MAP estimates by GP (left), Besov (middle) and Q-EP (right) models. (b)(d) Predictions by GP (left) and Q-EP (right) models. Orange dots are actual realizations (data points). Blue solid lines are true trajectories. Black ticks indicate the training data points. Red dashed lines are MAP estimates. Red dot-dashed lines are predictions with shaded region being credible bands.

We use the above likelihood and test three priors: GP, Besov and Q-EP. For Besov, we choose the Fourier basis $\phi_0(t) = \sqrt{2}$, $\phi_\ell(t) = \cos(\pi \ell t)$, $\ell \in \mathbb{N}$ (results with other wavelet bases including Haar, Shannon, Meyer and Mexican Hat are worse hence omitted). For both GP and Q-EP, we adopt the Matérn kernel with $\nu = \frac{1}{2}$, $\sigma^2 = 1$, $\rho = 0.5$ and $s = 1$: $C(t, t') = \sigma^2 \frac{2^{1-\nu}}{\Gamma(\nu)} w^\nu K_\nu(w)$, $w = \sqrt{2\nu}(\|t - t'\|/\rho)^s$. In both Besov and Q-EP, we set $q = 1$. Figures 4a and 4c (and Figures C.2a and C.2c) compare the MAP estimates (red dashed lines). We can see that Q-EP yields the best estimates closest to the true trajectories in the simulation and the best fit to the Tesla/Google stock prices. We also investigate the negative posterior densities and relative errors, $\|\hat{u}_* - u_*\|/\|u_*\|$, as functions of iterations in Figure C.1. Though incomparable in the absolute values, the negative posterior densities indicate faster convergence in both GP and Q-EP models than in Besov model. The error reducing plots on the right panels of subplots in Figure C.1 indicate that Q-EP prior model can achieve the smallest errors. Table 1 compares them in terms of root mean of squared error (RMSE) and log-likelihood (LL).

Table 1: Time series modeling: root mean of squared errors (RMSE) and log-likelihood (LL) values at MAP estimates by GP, Besov and Q-EP prior models.

| Data Sets | root mean squared errors (RMSE) | | | log-likelihood (LL) | | |
|---|---|---|---|---|---|---|
| | GP | Besov | Q-EP | GP | Besov | Q-EP |
| simulation (jumps) | 1.2702 | 2.1603 | **1.1083** | -31.4582 | -89.8549 | -74.0590 |
| simulation (turnings) | 1.4270 | 2.4556 | **0.9987** | -39.8234 | -56.7874 | -87.3124 |
| Tesla stocks | 180.3769 | 136.8769 | **51.2236** | -488.6458 | -281.3796 | -39.4070 |
| Google stocks | 44.4236 | 39.4809 | **36.8686** | -386.1546 | -305.0058 | -265.9790 |

Then we consider the prediction problem. In the simulations, the last $1/8$ portion and every other of the last but $3/8$ part of the data points are selected for testing. The models with GP and Q-EP priors are trained on the rest of the data, as indicated by short "ticks" in Figures 4b and 4d (and Figures C.2b and C.2d). For the Tesla/google stocks, we select every other day in the first half year, every 4 days in the 3rd quarter and every 8 days in the last quarter for training and test on the rest. They pose challenges on both interpolation (among observations) and extrapolation (at no-observation region) tasks. As we can see in those figures, uncertainty grows as the data become scarce. Nevertheless, the Q-EP yields smaller errors than GP. Note, such prediction is not immediately available for models with Besov prior.

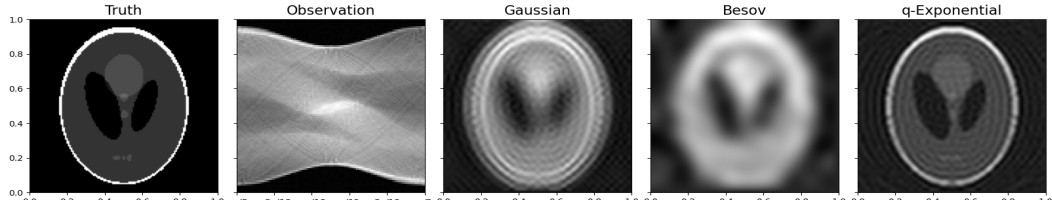

Figure 5: Shepp-Logan phantom: true image, observation (sinogram), and MAP estimates by GP, Besov and Q-EP models with relative errors 68.10%, 70.27% and **40.87%** respectively.

## 4.2 Computed Tomography Imaging

Computed tomography (CT) is a medical imaging technique used to obtain detailed internal images of human body. CT scanners use a rotating X-ray tube and a row of detectors to measure X-ray attenuations by different tissues inside the body from different angles. Denote the true imaging as a function $u(x)$ on the square unit $D = [0, 1]^2$ taking values as the pixels. The observed data, $\mathbf{y}$, (a.k.a. sinogram) are results of Radon transformation ($\mathbf{A}$) of the discretized $n \times n$ field $\mathbf{u}$ with $n_\theta$ angles and $n_s$ sensors, contaminated by noise $\boldsymbol{\varepsilon}$ [6]:

$$\mathbf{y} = \mathbf{Au} + \boldsymbol{\varepsilon}, \quad \boldsymbol{\varepsilon} \sim \mathscr{N}(\mathbf{0}, \sigma_\varepsilon^2 \mathbf{I}), \quad \mathbf{y} \in \mathbb{R}^{n_\theta n_s}, \quad \mathbf{A} \in \mathbb{R}^{n_\theta n_s \times n^2}, \quad \mathbf{u} \in \mathbb{R}^{n^2}$$

In general $n_\theta n_s \ll d = n^2$ so the linear inverse problem is under-determined. Baysian approach could fill useful prior information (e.g. edges) in the sparse data.

We first consider the Shepp–Logan phantom, a standard test image created by Shepp and Logan in [42] to model a human head and to test image reconstruction algorithms. In this simulation, we create the true image $u^\dagger$ for a resolution of $n^2 = 128 \times 128$ and project it at $n_\theta = 90$ angles with $n_s = 100$ equally spaced sensors. The generated sinogram is then added by noise with signal noise ratio $\text{SNR} = \|\mathbf{Au}^\dagger\| / \|\boldsymbol{\varepsilon}\| = 100$. The first two panels of Figure 5 show the truth and the observation.

Table 2: Posterior estimates of Shepp–Logan phantom by GP, Besov and Q-EP prior models: relative error, $\text{RLE} := \|\hat{u} - u^\dagger\| / \|u^\dagger\|$, of MAP ($\hat{u} = u^*$) and posterior mean ($\hat{u} = \bar{u}$) respectively, log-likelihood (LL), PSNR, SSIM and HarrPSI. Numbers in the bracket are standard deviations obtained repeating the experiments for 10 times with different random seeds.

| | MAP | | | Posterior Mean | | |
|---|---|---|---|---|---|---|
| | GP | Besov | Q-EP | GP | Besov | Q-EP |
| RLE | 0.6810 | 0.7027 | **0.4087** | 0.4917(6.16e-7) | 0.4894(3.53e-5) | **0.4890**(4.79e-5) |
| LL | -1.55e+6 | -1.54e+6 | -1.57e+5 | -5.21e+5(8.47) | -4.80e+5(196.34) | -4.56e+5(307.97) |
| PSNR | 15.5531 | 15.2806 | **19.9887** | 18.3826(1.09e-5) | 18.4226(6.27e-4) | **18.4303**(8.51e-4) |
| SSIM | 0.4028 | 0.3703 | **0.5967** | **0.5561**(3.92e-7) | 0.5535(2.38e-4) | 0.5403(5.26e-4) |
| HaarPSI | 0.0961 | 0.0870 | **0.3105** | 0.3126(1.52e-8) | **0.3126**(3.36e-4) | 0.3122(3.06e-4) |

Note, the computation involving a full sized ($d \times d$) kernel matrix $\mathbf{C}$ for GP and Q-EP is prohibitive. Therefore, we consider its Mercer's expansion (12) with Fourier basis for a truncation at the first $L = 2000$ items. Figure 5 shows that while GP and Besov models reconstruct very blurry phantom images, the Q-EP prior model produces MAP estimate of the highest quality. For each of the three models, we also apply wn-pCN to generate 10000 posterior samples (after discarding 5000) and use them to reconstruct $u$ (posterior mean or median) and quantify uncertainty (posterior standard deviation).

Table 2 summarizes the errors relative to MAP ($u^*$) and posterior mean ($\bar{u}$) respectively, $\|\hat{u} - u^\dagger\| / \|u^\dagger\|$ (with $\hat{u}$ being $u^*$ or $\bar{u}$), log-likelihood (LL), and several quality metrics in imaging analysis including the peak signal-to-noise ratio (PSNR) [20], the structured similarity index (SSIM) [46], and the Haar wavelet-based perceptual similarity index (HaarPSI) [39]. Q-EP attains the lowest error and highest quality scores in most cases. In Figure C.3, we compare the uncertainty by these models. It seems that GP has uncertainty filed with more recognizable shape than the other two. However, the posterior standard deviation by GP is much smaller (about 1% of that with Q-EP) compared with the other two. Therefore, this raises a red flag that GP could be over-confident about a less accurate estimate.

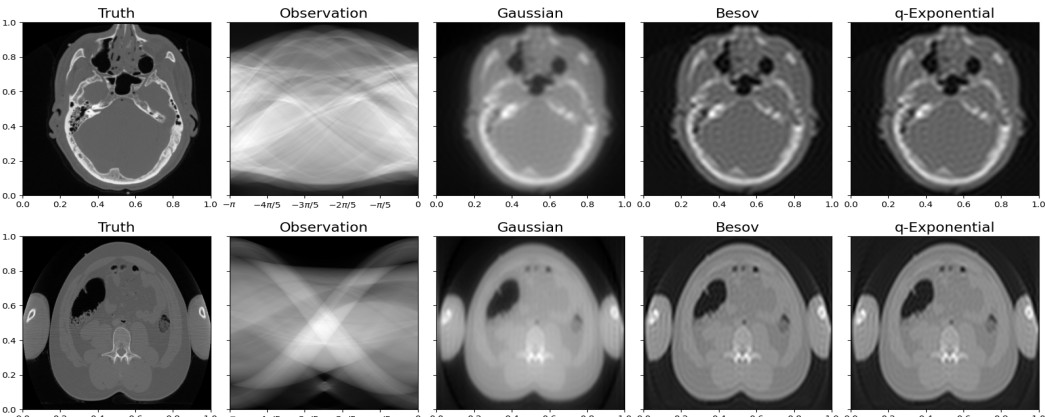

Figure 6: CT of human head (upper) and torso (lower): true image, observation (sinogram), and MAP estimates by GP, Besov and Q-EP models with relative errors 29.99%, 22.41% and **22.24%** (for head) and 26.11%, 21.77% and **21.53%** (for torso) respectively.

Finally, we apply these methods to CT scans of a human cadaver and torso from the Visible Human Project [1]. These images contain $n^2 = 512 \times 512$ pixels and the sinograms are obtained with $n_\theta = 200$ angles and $n_s = 512$ sensors. The first two panels of each row in Figure 6 show a highly calibrated CT reconstruction (treated as "truth") and the observed sinogram. The rest three panels illustrate that both Besov and Q-EP models outperform GP in reconstructions, as verified in the quantitative summaries in Table C.2. Figure C.4 indicates that GP tends to underestimate the uncertainty.

In these CT reconstruction examples, we observe larger discrepancy of performance between Besov and Q-EP in the low-dimensional data-sparse application (Shepp–Logan phantom at resolution $n^2 = 128 \times 128$ with $n_\theta = 90$ angles and $n_s = 100$ sensors) compared with the high-dimensional data-intensive applications (two human body CTs at resolution $n^2 = 512 \times 512$ with $n_\theta = 200$ angles and $n_s = 512$ sensors). This may be due to the truncation in Mercer's kernel representation (12) and different rates of posterior contraction [21, 22, 2]. We will explore them in another journal paper.

## 5 CONCLUSION

In this paper, we propose the $q$-exponential process (Q-EP) as a prior on $L^q$ functions with a flexible parameter $q > 0$ to control the degree of regularization. Usually, $q = 1$ is adopted to capture abrupt changes or sharp contrast in data such as edges in the image as the Besov prior has recently gained popularity for. Compared with GP, Q-EP can impose sharper regularization through $q$. Compared with Besov, Q-EP enjoys the explicit formula with more control on the correlation structure as GP. The numerical experiments in time series modeling, image reconstruction and Bayesian inverse problems demonstrate our proposed Q-EP is superior in Bayesian functional data modeling.

In the numerical experiments of current work, we manually grid-search for the optimal hyper-parameters. The reported results are not sensitive to some of these hyper-parameters such as the variance magnitude ($\sigma^2$) and the correlation length ($\rho$) but may change drastically to others like the regularity parameter ($\nu$) and the smoothness parameter ($s$). In future, we will incorporate hyper-priors for some of those parameters and adopt a hierarchical scheme to overcome such shortcoming. We plan to study the properties such as regularity of function draws of Q-EP and the posterior contraction, and compare the contraction rates among GP, Besov and Q-EP priors [21, 22, 2]. Future work will also consider operator based kernels such as graph Laplacian [16, 17, 29].

## Acknowledgments and Disclosure of Funding

SL is supported by NSF grant DMS-2134256.

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

# Supplement Document for "Bayesian Learning via Q-Exponential Process"

## A  PROOFS

### A.1  Proof of Theorem 3.3

*Proof.* First we prove the exchangeability of $q-\mathrm{ED}_d(\boldsymbol{\mu}, \mathbf{C})$ with general (non-identity) covariance matrix $\mathbf{C} = [\mathscr{C}(t_i, t_j)]_{d \times d}$ for some kernel function $\mathscr{C}$. It actually holds for all elliptic distributions including MVN. Their densities contain the essential quadratic form $r(\mathbf{u}) = \mathbf{u}^\mathsf{T} \mathbf{C}^{-1} \mathbf{u}$ which is invariant under any permutation of coordinates.

Denote $\mathbf{u} = [u_{t_1}, \cdots, u_{t_i}, \cdots, u_{t_j}, \cdots, u_{t_d}]^T$. Without loss of generality, we only need to show $r(\mathbf{u})$ is invariant by switching two coordinates, say, $t_i \leftrightarrow t_j$. Denote $\mathbf{u}' = [u_{t_1}, \cdots, u_{t_j}, \cdots, u_{t_i}, \cdots, u_{t_d}]^T$. Switching $t_i$ and $t_j$ leads to a different covariance matrix $\mathbf{C}'$ obtained by switching both $i$-th and $j$-th rows and columns simultaneously in $\mathbf{C}$. If we denote the elementary matrix $E_{ij}$ as derived from switching $i$-th and $j$-th rows of the identity matrix $\mathbf{I}$. Then we have

$$\mathbf{u}' = E_{ij}\mathbf{u}, \quad \mathbf{C}' = E_{ij}\mathbf{C}E_{ij}$$

Note $E_{ij}$ is idempotent, i.e. $E_{ij} = E_{ij}^{-1}$. Therefore

$$(\mathbf{u}')^\mathsf{T}(\mathbf{C}')^{-1}\mathbf{u}' = \mathbf{u}^\mathsf{T} E_{ij}E_{ij}\mathbf{C}^{-1}E_{ij}E_{ij}\mathbf{u} = \mathbf{u}^\mathsf{T}\mathbf{C}^{-1}\mathbf{u}$$

Next, the consistency directly follows from Kano's consistency Theorem 3.2 with our choice of $g(r)$. The proof is hence completed. $\square$

### A.2  Theorem of Q-EP as a mixture of Gaussians

**Theorem A.1.** *Suppose* $\mathbf{u} \sim q-\mathrm{ED}_d(0, \mathbf{C})$ *for* $0 < q < 2$*, then there exist an random variable* $V > 0$ *and a standard normal random vector* $\mathbf{Z} \sim \mathscr{N}_d(\mathbf{0}, \mathbf{I})$ *independent of each other such that* $\mathbf{u} \overset{d}{=} \mathbf{Z}/V$.

*Proof.* Based on [3], it suffices to show $(-\frac{d}{dr})^k g(r) \geq 0$ for all $k \in \mathbb{N}$. Observe that $g'(r) = \left[ (\frac{q}{2} - 1)\frac{d}{2}r^{(\frac{q}{2}-1)\frac{d}{2}-1} - \frac{q}{4}r^{(\frac{q}{2}-1)(\frac{d}{2}+1)} \right] \exp\{-\frac{r^{\frac{q}{2}}}{2}\} \leq 0$ when $q \leq 2$. Denote $(-\frac{d}{dr})^k g(r) := p_k(r^{(\frac{q}{2}-1)/2}, r^{-1}) \exp\{-\frac{r^{\frac{q}{2}}}{2}\}$ where the coefficients of polynomial $p_k$ are all non-negative. Then we have

$$\left( -\frac{d}{dr} \right)^{k+1} g(r) = \left[ -\frac{d}{dr}p_k(r^{(\frac{q}{2}-1)/2}, r^{-1}) + \frac{q}{4}r^{(\frac{q}{2}-1)}p_k(r^{(\frac{q}{2}-1)/2}, r^{-1}) \right] \exp\{-\frac{r^{\frac{q}{2}}}{2}\}$$

where $p_{k+1}(r^{(\frac{q}{2}-1)/2}, r^{-1})$ being the term in the square bracket has all positive coefficients because the powers $(\frac{q}{2} - 1)/2$ and $-1$ appear as coefficients in $\frac{d}{dr}p_k(r^{(\frac{q}{2}-1)/2}, r^{-1})$ and are both negative. The proof is completed by induction. $\square$

### A.3  Proposition of distribution of $r(\mathbf{u})$

The following proposition determines the distribution of $R = \sqrt{r(\mathbf{u})}$ as $q$-root of a gamma (also *chi*-squared) distribution thus gives a complete recipe for generating random vector $\mathbf{u} \sim q-\mathrm{ED}_d(0, \mathbf{C})$ based on the stochastic representation (6).

**Proposition A.1.** *If* $\mathbf{u} \sim q-\mathrm{ED}_d(0, \mathbf{C})$*, then we have*

$$R^q = r^{\frac{q}{2}} \sim \Gamma\left( \alpha = \frac{d}{2}, \beta = \frac{1}{2} \right) = \chi_d^2, \quad and \quad \mathrm{E}[R^k] = 2^{\frac{k}{q}}\frac{\Gamma(\frac{d}{2} + \frac{k}{q})}{\Gamma(\frac{d}{2})} \overset{\cdot}{\sim} d^{\frac{k}{q}}, \; as \; d \to \infty, \; \forall k \in \mathbb{N} \quad (18)$$

*Proof.* With out chosen $g(r)$, the density of $r$ becomes

$$f(r) \propto r^{\frac{d}{2}-1}r^{(\frac{q}{2}-1)\frac{d}{2}} \exp\left\{ -\frac{r^{\frac{q}{2}}}{2} \right\} = r^{\frac{q}{2} \cdot \frac{d}{2}-1} \exp\left\{ -\frac{r^{\frac{q}{2}}}{2} \right\}$$

A change of variable $r \to r^{\frac{q}{2}}$ yields the density of $R^q = r^{\frac{q}{2}}$ that can be recognized as the density of $\chi_d^2$.

On the other hand, since $v := R^q \sim \Gamma\left(\alpha = \frac{d}{2}, \beta = \frac{1}{2}\right)$, we have:

$$\mathrm{E}[R^k] = \int_0^\infty v^{\frac{k}{q}} f(v) dv = \frac{1}{\Gamma(\frac{d}{2})} \left(\frac{1}{2}\right)^{\frac{d}{2}} \int_0^\infty v^{\frac{k}{q} + \frac{d}{2} - 1} \exp\left\{-\frac{1}{2}v\right\} dv$$

$$= 2^{\frac{k}{q}} \frac{\Gamma(\frac{d}{2} + \frac{k}{q})}{\Gamma(\frac{d}{2})} \overset{.}{\sim} 2^{\frac{k}{q}} \left(\frac{d}{2}\right)^{\frac{k}{q}} = d^{\frac{k}{q}}$$

where we use $\Gamma(x + \alpha) \overset{.}{\sim} \Gamma(x) x^\alpha$ as $x \to \infty$ with $x = \frac{d}{2}$ and $\alpha = \frac{k}{q}$ when $d \to \infty$. $\qquad\square$

## A.4 Proof of Proposition 3.1

*Proof.* By Theorem 2.6.4 in [19] for $q-\mathrm{ED}_d(\boldsymbol{\mu}, \mathbf{C}) = \mathrm{EC}_d(\boldsymbol{\mu}, \mathbf{C}, g)$ with our chosen $g$, we know $\mathrm{E}[\mathbf{u}] = \boldsymbol{\mu}$ and $\mathrm{Cov}(\mathbf{u}) = (\mathrm{E}[R^2]/\mathrm{rank}(\mathbf{C}))\mathbf{C}$. It follows by letting $k = 2$ in Proposition A.1 and using the similar asymptotic analysis. $\qquad\square$

## A.5 Proof of Theorem 3.4

*Proof.* Note we can approximate $\phi_\ell(x) \in L^2(D)$ with simple functions $\tilde{\phi}_\ell(x) = \sum_{i=1}^d k_i \chi_{D_i}(x)$ where $D_i$'s are measurable subsets of $D$ and $\chi_{D_i}(x) = 1$ if $x \in D_i$ and 0 otherwise. By the linear combination property of elliptic distributions [c.f. Theorem 2.6.3 in 19], $\tilde{u}_\ell = \int_D u(x)\tilde{\phi}_\ell(x)dx \sim q-\mathrm{ED}(0, c)$ with $c = \alpha_d^{-1}\mathrm{E}[\tilde{u}_\ell^2]$ to be determined. Note $\alpha_d = \frac{2^{\frac{2}{q}}\Gamma(\frac{d}{2} + \frac{2}{q})}{d\Gamma(\frac{d}{2})} d^{1-\frac{2}{q}}$ comes from Proposition 3.1 and the scaling $\mathbf{u}^* = d^{\frac{1}{2} - \frac{1}{q}}\mathbf{u}$ in Definition 3.2. We have $\alpha_d = \frac{\Gamma(\frac{d}{2} + \frac{2}{q})}{\Gamma(\frac{d}{2})}\left(\frac{2}{d}\right)^{\frac{2}{q}} \to 1$ as $d \to \infty$. Taking the limit $d \to \infty$, we have $u_\ell = \int_D u(x)\phi_\ell(x)dx \sim q-\mathrm{ED}(0, c)$. In general, by the similar argument we have

$$\mathrm{Cov}(u_\ell, u_{\ell'}) = \mathrm{E}[u_\ell u_{\ell'}] = \int_D \int_D \mathrm{E}[u(x)u(x')]\phi_\ell(x)\phi_{\ell'}(x')dxdx'$$

$$= \int_D \int_D \mathscr{C}(x, x')\phi_\ell(x)\phi_{\ell'}(x')dxdx' = \int_D \lambda_\ell \phi_\ell(x')\phi_{\ell'}(x')dx' = \lambda_\ell \delta_{\ell\ell'}$$

Thus it completes the proof. $\qquad\square$

## A.6 Proof of Theorem 3.5

Before proving Theorem 3.5, we first prove the following lemma based on the conditional of elliptic distribution [12, 19].

**Lemma A.1.** *If* $\mathbf{u} = (\mathbf{u}_1, \mathbf{u}_2) \sim q-\mathrm{ED}_d(\boldsymbol{\mu}, \mathbf{C})$ *with* $\boldsymbol{\mu} = \begin{bmatrix} \boldsymbol{\mu}_1 \\ \boldsymbol{\mu}_2 \end{bmatrix}$ *and* $\mathbf{C} = \begin{bmatrix} \mathbf{C}_{11} & \mathbf{C}_{12} \\ \mathbf{C}_{21} & \mathbf{C}_{22} \end{bmatrix}$, $\mathbf{u} \in \mathbb{R}^d$, $\mathbf{u}_i \in \mathbb{R}^{d_i}$ *for* $i = 1, 2$ *and* $d_1 + d_2 = d$, *then we have the following conditional distribution*

$$\mathbf{u}_1 | \mathbf{u}_2 \sim q-\mathrm{ED}_{d_1}(\boldsymbol{\mu}_{1\cdot2}, \mathbf{C}_{11\cdot2}),$$
$$\boldsymbol{\mu}_{1\cdot2} = \boldsymbol{\mu}_1 + \mathbf{C}_{12}\mathbf{C}_{22}^{-1}(\mathbf{u}_2 - \boldsymbol{\mu}_2), \quad \mathbf{C}_{11\cdot2} = \mathbf{C}_{11} - \mathbf{C}_{12}\mathbf{C}_{22}^{-1}\mathbf{C}_{21}$$

*Proof.* This directly follows from [Corollary 5 of Theorem 5 in 12] or [Corollary 3 of Theorem 2.6.6 in 19] for $q-\mathrm{ED}_d(\boldsymbol{\mu}, \mathbf{C}) = \mathrm{EC}_d(\boldsymbol{\mu}, \mathbf{C}, g)$ with our chosen $g$. $\qquad\square$

Now we prove the Theorem 3.5.

*Proof.* By the linear combination property of the elliptic distributions [26, 19], we have $\mathbf{y} \sim q-\mathrm{ED}(\mathbf{0}, \mathbf{C} + \Sigma)$. Then based on the consistency, we have the joint distribution

$$\begin{bmatrix} \mathbf{y} \\ u(x_*) \end{bmatrix} \sim q-\mathrm{ED}\left(\mathbf{0}, \begin{bmatrix} \mathbf{C} + \Sigma & \mathbf{C}_* \\ \mathbf{C}_*^\mathsf{T} & \mathbf{C}_{**} \end{bmatrix}\right)$$

Therefore, the conclusion follows from Lemma A.1. $\qquad\square$

## A.7 Proposition of Conditional Conjugacy for Variance Magnitude ($\sigma^2$)

**Proposition A.2.** *If we assume a proper inverse-gamma prior for the variance magnitude such that* $\mathbf{u}|\sigma^2 \sim \mathrm{q-ED}_d(\boldsymbol{\mu}, \mathbf{C} = \sigma^2 \mathbf{C}_0)$, *and* $\sigma^q \sim \Gamma^{-1}(\alpha, \beta)$, *then we have*

$$\sigma^q|\mathbf{u} \sim \Gamma^{-1}(\alpha', \beta'), \quad \alpha' = \alpha + \frac{d}{2}, \quad \beta' = \beta + \frac{(\mathbf{u} - \boldsymbol{\mu})^\mathsf{T} \mathbf{C}_0^{-1}(\mathbf{u} - \boldsymbol{\mu})}{2} \tag{19}$$

*Proof.* Denote $r_0 = (\mathbf{u} - \boldsymbol{\mu})^\mathsf{T} \mathbf{C}_0^{-1}(\mathbf{u} - \boldsymbol{\mu})$. We can compute the joint density of $\mathbf{u}$ and $\sigma^2$

$$\begin{aligned}
p(\mathbf{u}, \sigma^2) &= p(\mathbf{u}|\sigma^2)p(\sigma^q) \\
&= \frac{q}{2}(2\pi)^{-\frac{d}{2}}|\mathbf{C}_0|^{-\frac{1}{2}} r_0^{(\frac{q}{2}-1)\frac{d}{2}} \sigma^{-\frac{qd}{2}} \exp\left\{-\sigma^{-q}\frac{r_0^{\frac{q}{2}}}{2}\right\} \frac{\beta^\alpha}{\Gamma(\alpha)}(\sigma^q)^{-(\alpha+1)} \exp(-\beta\sigma^{-q}) \\
&\propto (\sigma^q)^{-(\alpha+\frac{d}{2}+1)} \exp\left\{-\sigma^{-q}\left(\beta + \frac{r_0^q}{2}\right)\right\}
\end{aligned}$$

By identifying the parameters for $\sigma^q$ we recognize that $\sigma^q|\mathbf{u}$ is another inverse-gamma with parameters $\alpha'$ and $\beta'$ as given. $\qquad\square$

# B ALGORITHM

---

**Algorithm 1** White-noise Preconditioed Crank-Nicolson (wn-pCN) for Q-EP Prior Models

---

1: Fix $\beta \in (0, 1]$. Choose initial state $\mathbf{z}^{(0)} \in \mathbb{R}^d$.
2: **for** $k = 0, \cdots, K-1$ **do**
3:     Propose $\hat{\mathbf{z}}^{(k)} = (1 - \beta^2)^{\frac{1}{2}}\mathbf{z}^{(k)} + \beta\mathbf{z}'$, $\mathbf{z}' \sim \mathcal{N}(\mathbf{0}, \mathbf{I})$.
4:     Set $\mathbf{z}^{(k+1)} = \hat{\mathbf{z}}^{(k)}$ with acceptance probability

$$\min\left\{1, \frac{L(T(\hat{\mathbf{z}}^{(k)}))|dT(\hat{\mathbf{z}}^{(k)})|}{L(T(\mathbf{z}^{(k)}))|dT(\mathbf{z}^{(k)})|}\right\}$$

5:     or else set $\mathbf{z}^{(k+1)} = \mathbf{z}^{(k)}$.
6: **end for**

---

# C ADDITIONAL EXPERIMENTAL RESULTS

In this section, we present some additional numerical experimental results that cannot be included in the main text due to the page limit.

First, we numerically verify the equivalence between the stochastic representation (6) and the white-noise representation (17) of $\mathrm{q-ED}_d$ random variable in Figure B.1. More specifically, we generate 10000 samples using each of these two representations and illustrate in Figure B.1 that the two samples yield empirical marginal distributions (1d and 2d) close enough to each other.

## C.1 Time Series Modeling

For modeling the simulated time series and stock prices, we include the optimization trace of negative (log)-posterior densities and relative errors for the two simulations and two stocks prices in Figure C.1. As commented in the main text, these plots show that Q-EP model can converge faster to lower errors compared with GP and Besov models.

Next, we compare MAP estimates by GP, Besov and Q-EP models in Figure C.2a for simulated time series with step jumps and in Figure C.2c for the Google stock prices in 2022. We also investigate the prediction results by GP and Q-EP in these two examples in Figures C.2b and C.2d. Table C.1 summarizes the RMSE of estimated stock prices by the three models and its standard deviation for repeating the experiments 10 times independently.

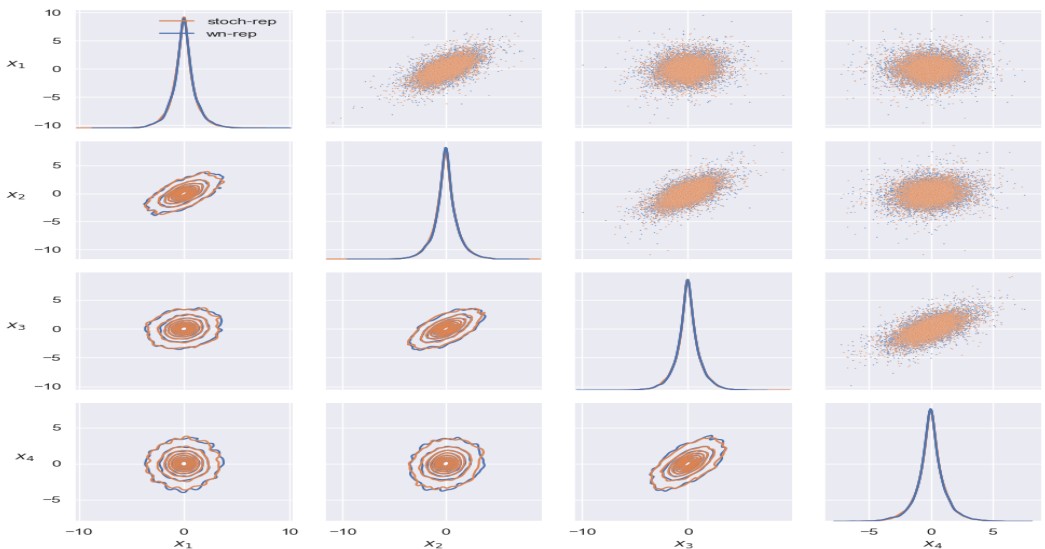

Figure B.1: Comparison in sampling q$-$ED$_d$ using the stochastic representation (6) (organge) and the white-noise representation (17) (blue). Numerical results show their sampling distributions are indistinguishable. Empirical densities are estimated based on 10000 samples (shown as dots).

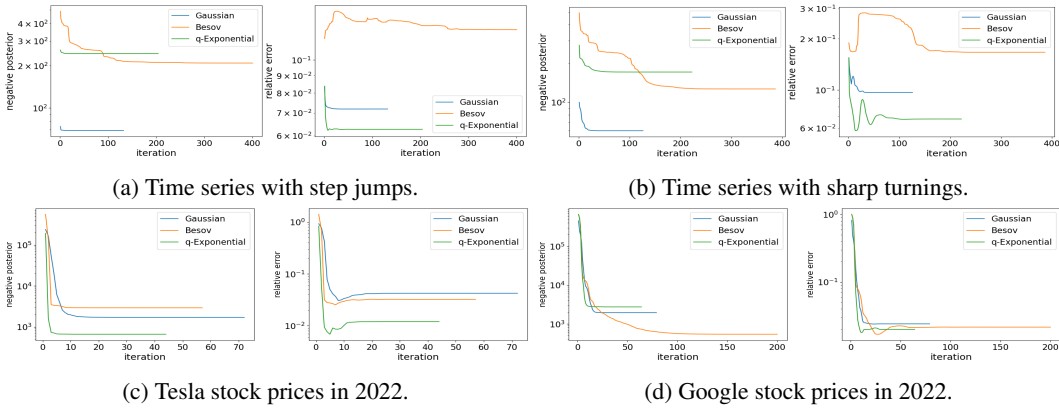

(a) Time series with step jumps.

(b) Time series with sharp turnings.

(c) Tesla stock prices in 2022.

(d) Google stock prices in 2022.

Figure C.1: Negative posterior densities (left) and errors (right) as functions of iterations in the BFGS algorithm used to obtain MAP estimates. Early termination is implemented if the error falls below some threshold or the maximal iteration (1000) is reached. Relative errors are compared against truth in the simulation and the actual data in the Tesla stock.

## C.2 Computed Tomography Imaging

In the problem of reconstructing human head and torso CT images, Table C.2 compares GP, Besov and Q-EP models in terms of relative error (RLE), log-likelihood (LL), and imaging quality metrics including PSNR, SSIM and HarrPSI. In most cases, Q-EP outperforms, or achieves comparable scores with the other two methods.

Lastly, Figures C.3 and C.4 show that the posterior standard deviations estimated by wn-pCN using GP model could be misleading because the seemingly more recognizable shape deludes the fact that they are about two orders of magnitude smaller in value compared with the other two models. This implies that GP might underestimate the uncertainty present in the observed sinograms in the CT imaging analysis.

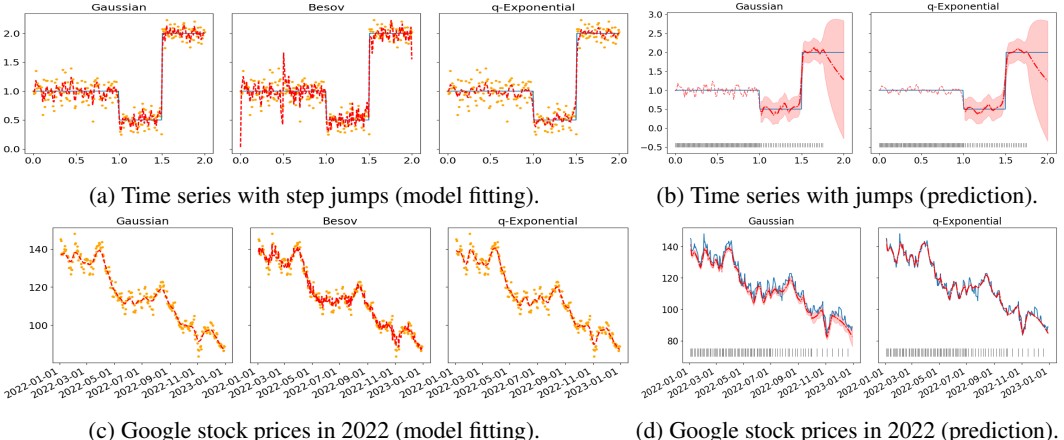

(a) Time series with step jumps (model fitting).  (b) Time series with jumps (prediction).

(c) Google stock prices in 2022 (model fitting).  (d) Google stock prices in 2022 (prediction).

Figure C.2: (a)(c) MAP estimates by GP (left), Besov (middle) and Q-EP (right) models. (b)(d) Predictions by GP (left) and Q-EP (right) models. Orange dots are actual realizations (data points). Blue solid lines are true trajectories. Black ticks indicate the training data points. Red dashed lines are MAP estimates. Red dot-dashed lines are predictions with shaded region being credible bands.



Figure C.3: Shepp–Logan phantom: uncertainty field (posterior standard deviation) given by GP, Besov and Q-EP models. GP tends to underestimate the uncertainty values (about 1% of that with Q-EP).

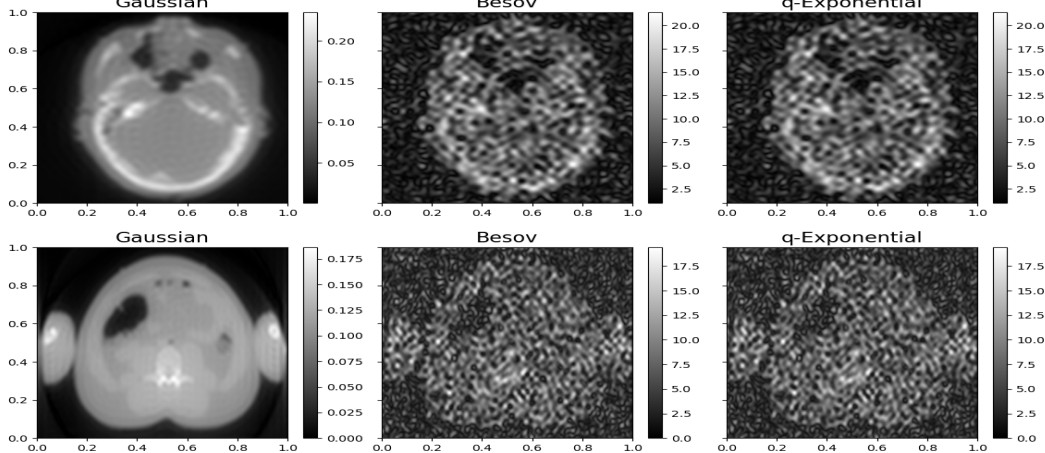

Figure C.4: CT of human head (upper) and torso (lower): uncertainty field (posterior standard deviation) given by GP, Besov and Q-EP models. Note GP tends to underestimate the uncertainty values (about 1% of that with Q-EP).

Table C.1: Posterior estimates of Tesla and Google stock prices by GP, Besov and Q-EP prior models: RMSE $:= \|\bar{u} - u\|_2$. Results are repeated 10 times with different random seeds.

|  | Tesla | | | Google | | |
|---|---|---|---|---|---|---|
|  | GP | Besov | Q-EP | GP | Besov | Q-EP |
| RMSE | 171.8515 | 90.3086 | **83.8130** | 20.4095 | 25.2012 | **18.3597** |
| std(RMSE) | 1.8018 | 1.1478 | 2.6949 | 0.7115 | 0.1698 | 0.9617 |

Table C.2: MAP estimates for CT of human head and torso by GP, Besov and Q-EP prior models: relative error, RLE $:= \|\hat{u} - u^\dagger\|/\|u^\dagger\|$ of MAP ($\hat{u} = u^*$), log-likelihood (LL), PSNR, SSIM and HarrPSI.

|  | Head | | | Torso | | |
|---|---|---|---|---|---|---|
|  | GP | Besov | Q-EP | GP | Besov | Q-EP |
| RLE | 0.2999 | 0.2241 | **0.2224** | 0.2611 | 0.2177 | **0.2153** |
| LL | -4.05e+5 | -1.12e+4 | -1.17e+4 | -3.30e+5 | -3.86e+3 | -4.37e+3 |
| PSNR | 24.2321 | 26.7633 | **26.8281** | 23.6450 | 25.2231 | **25.3190** |
| SSIM | 0.7010 | 0.7914 | **0.8096** | 0.5852 | **0.6983** | 0.6982 |
| HaarPSI | 0.0525 | **0.0593** | 0.0587 | 0.0666 | **0.0732** | 0.07190 |

### C.3 Noisy/Blurry Image Reconstruction

Next we consider reconstructing a ($128 \times 128$ pixels) image of satellite shown on the leftmost of Figure 1 from a blurred observation next to it. The image itself can be viewed as a function $u(x)$ on the square unit $D = [0,1]^2$ taking values as the pixels. When evaluating $u(x)$ on the discretized domain, $u(x)$ becomes a matrix of size $128 \times 128$, which can further be vectorized to $\mathbf{u} \in \mathbb{R}^d$ with $d = 128^2$. The true image, denoted as $u^\dagger$, is blurred by applying a motion blur point spread function [PSF 11] and adding 5% Gaussian noise. The actual observation, $y(x)$, can be written as in the following linear model:

$$y(x) = Au(x) + \varepsilon, \quad \varepsilon \sim \mathcal{N}(0, \sigma_\varepsilon^2 I_d)$$

where $A \in \mathbb{R}^{J \times d}$ is the blur motion PSF with $J = d$ and $\sigma_\varepsilon / \|Au\| = 5\%$. Note, the blurring effect in the observed image (the second from left of Figure 1) is mainly due to the PSF operator $A$, not the small Gaussian noise.

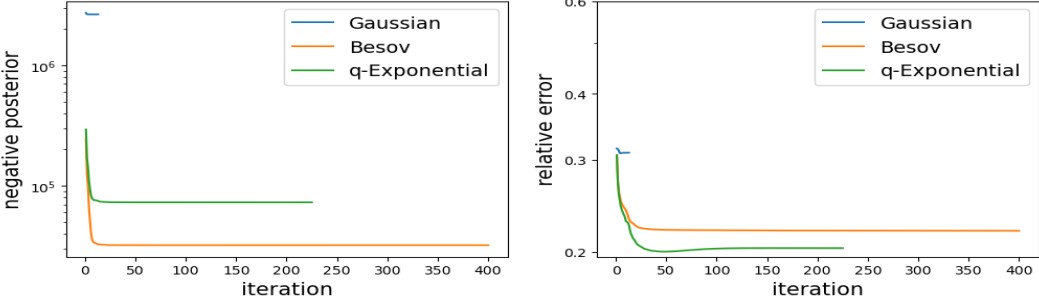

Figure C.5: Image of satellite: negative posterior densities (left) and errors (right) as functions of iterations in the BFGS algorithm used to obtain MAP estimates. Early termination is implemented if the error falls below some threshold or the maximal iteration (1000) is reached.

We compare the reconstructions by MAP estimate in Figure 1. The output by GP is blurry and close to the observed image, which means that GP does not "de-noise" much. The result by Besov is much better than GP due to the $L_1$ regularization but it is still not sharp enough. We can see that the Q-EP prior model produces the reconstruction of the highest quality. Figure 3 demonstrates the effect of $q > 0$: the smaller $q$, the more regularization and hence sharper reconstruction. We also compare their negative posterior densities and relative errors, $\|\hat{u} - u^\dagger\|/\|u^\dagger\|$, in Figure C.5. The Q-EP prior model yields the smallest error among all the three models.

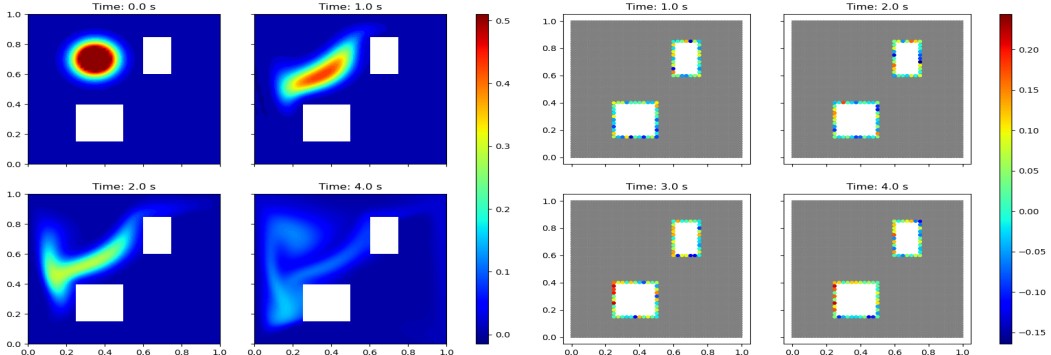

(a) True initial condition (top left), and the solutions $u(\mathbf{x},t)$ at different time points $t$.

(b) Spatiotemporal observations at 80 selected locations (color dots) across different time points.

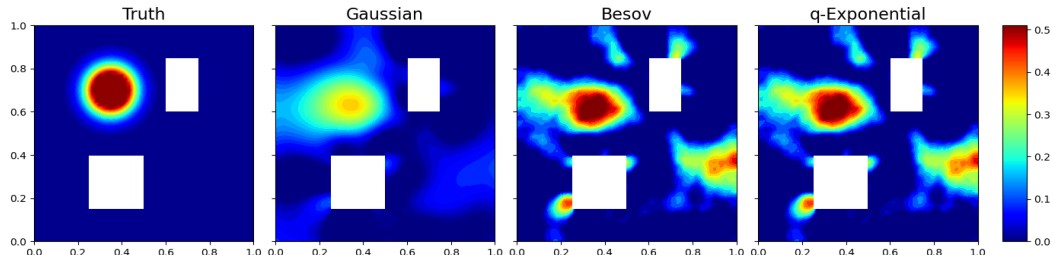

Figure C.7: Advection-diffusion inverse problem: true initial condition $u_0^\dagger$ and posterior mean estimates by GP, Besov and Q-EP models.

## C.4 Advection-Diffusion Inverse Problem

Finally, we consider a Bayesian inverse problem governed by a time-dependent advection-diffusion equation [36, 30] that can be applied to heat transfer, pollution tracing, etc. The inverse problem involves inferring an unknown initial condition $u_0 \in L^2(\Omega)$ from spatiotemporal point measurements $\{y(\mathbf{x}_i, t_j)\}$ as

$$y(\mathbf{x},t) = \mathscr{G}(u_0) + \eta(\mathbf{x},t), \quad \eta(\mathbf{x},t) \sim \mathscr{N}(0,\Sigma)$$

The forward mapping $\mathscr{G} : u_0 \to \mathscr{O}u$ maps the initial condition $u_0$ to pointwise spatiotemporal observations of the concentration field $u(\mathbf{x},t)$ through the solution of the following advection-diffusion equation [36, 45]:

$$u_t - \kappa\Delta u + \mathbf{v}\cdot\nabla u = 0 \quad in\ \Omega \times (0,T) \qquad -\frac{1}{\mathrm{Re}}\Delta\mathbf{v} + \nabla p + \mathbf{v}\cdot\nabla\mathbf{v} = 0 \quad in\ \Omega$$

$$u(\cdot,0) = u_0 \quad in\ \Omega \qquad\qquad\qquad \nabla\cdot\mathbf{v} = 0 \quad in\ \Omega$$

$$\kappa\nabla u\cdot\vec{n} = 0, \quad on\ \partial\Omega\times(0,T) \qquad\qquad \mathbf{v} = \mathbf{g}, \quad on\ \partial\Omega$$

where $\Omega \subset [0,1]^2$ is a bounded domain shown in Figure C.6a, $\kappa = 10^{-3}$ is the diffusion coefficient, and $T > 0$ is the final time. The velocity field $\mathbf{v}$ is computed by solving the following steady-state Navier-Stokes equation with the side walls driving the flow [36]. Here, $p$ is the pressure, and Re is the Reynolds number, which is set to 100 in this example. The Dirichlet boundary data $\mathbf{g} \in \mathbb{R}^2$ is given by $\mathbf{g} = \mathbf{e}_2 = (0,1)$ on the left wall, $\mathbf{g} = -\mathbf{e}_2$ on the right wall, and $\mathbf{g} = \mathbf{0}$ everywhere else.

To generate data, we set the true value of parameter $u_0$ in (C.4) as $u_0^\dagger = 0.5 \wedge \exp\{-100[(x-0.35)^2 + (y-0.7)^2]\}$, illustrated in the top left panel of Figure C.6a, which also shows a few snapshots of the solutions $u(\mathbf{x},t)$ at other time points on a regular grid mesh of size $61 \times 61$. Spatiotemporal observations $\{y(\mathbf{x}_i,t_j)\}_{i=1,j=1}^{I,J}$ are collected at $I = 80$ selected locations $\{\mathbf{x}_i\}_{i=1}^{I}$ around the boundary of two inner boxes (See Figure C.6a and also Figure C.6b ) across $J = 16$ time points $\{t_j\}_{j=1}^{J}$ evenly

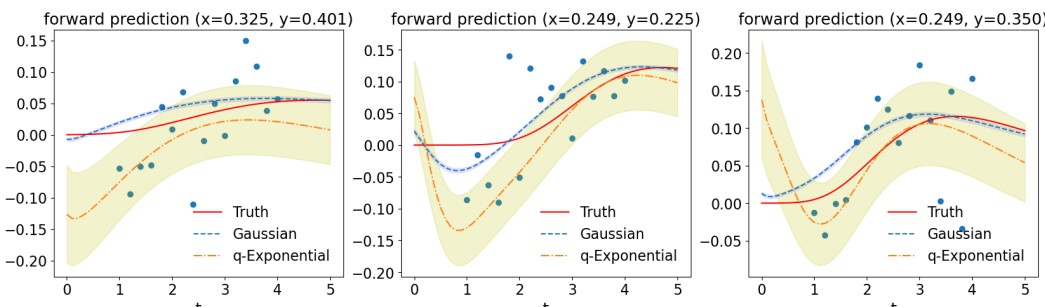

Figure C.8: Advection-diffusion inverse problem: comparing forward predictions, $\overline{\mathscr{G}}(\mathbf{x}, t_*)$, based on the GP (blue dashed lines) and Q-EP (orange dot-dashed lines) prior models at three selective locations $\mathbf{x} = (0.325, 0.401)$, $\mathbf{x} = (0.249, 0.225)$ and $\mathbf{x} = (0.249, 0.350)$. Blues dots are observations.

distributed between 1 and 4 seconds (thus denoted as $\mathscr{O}u$) with noise variance $\Sigma = \sigma_\eta^2 I_{1280}$ where $\sigma_\eta = 0.5 \max \mathscr{O}u$, i.e. $y(\mathbf{x}_i, t_j) = \mathscr{G}(u_0^\dagger) + \eta_{ij} = u(\mathbf{x}_i, t_j) + \eta_{ij}$.

To solve the inverse problem of finding the initial condition $u_0$ in the Bayesian framework [43, 16], we impose $u_0$ with GP, Besov and Q-EP priors respectively and seek the posterior $p(u_0|y)$. For GP and Q-EP, we adopt a covariance kernel, $\mathscr{C} = (\delta \mathscr{I} - \gamma \Delta)^{-2}$, defined through the Laplace operator $\Delta$, where $\delta$ governs the variance of the prior and $\gamma/\delta$ controls the correlation length [16, 30]. We set $\gamma = 1$ and $\delta = 8$ in this example. For Besov, we adopt 2d Fourier basis of the format $\phi_{ij} = \cos(\pi i x_1) \cos(\pi j x_2)$ and truncate the series (1) for the first $L = 1000$ terms.

We apply wn-pCN to this challenging nonlinear inverse problem with high dimensionality (3413) of spatially discretized $u$ at each time $t$. Figure C.7 compares the posterior mean estimates of $u_0$ given by these three models. Because the truth (leftmost) has clear edge at its cutoff by 0.5, Q-EP is more appropriate than GP and it indeed generates better estimate closer to the truth. Figure C.8 plots the prediction of forward mapping at a few selective locations on the left side of lower inner box by $\overline{\mathscr{G}}(\mathbf{x}, t_*) = \frac{1}{S} \sum_{s=1}^{S} \mathscr{G}(u^{(s)})(\mathbf{x}, t_*)$ with $u^{(s)} \sim p(u_0|y)$. Compared with GP, Q-EP predicts the solution path closer to the truth $\mathscr{G}(u_0^\dagger)(\mathbf{x}, t_*)$ where the observations see more dynamical changes. More importantly, Q-EP provides proper UQ with credible bands wide enough to include the true trajectories. On the other hand, the posterior estimates by GP come with much narrower error bands that miss the truth. Again, we observe GP prior model being overconfident about less accurate estimates.

