# OpenReview forum: "Bayesian Learning via Q-Exponential Process"
_NeurIPS.cc/2023/Conference — NeurIPS 2023 poster_

### Official Review · Reviewer_n1kr · 2023-07-06

**Soundness:** 3 good
**Presentation:** 4 excellent
**Contribution:** 3 good
**Rating:** 7
**Confidence:** 2

**Summary:**

This paper proposes a generalization of Besov processes to higgher dimensions that satisfies the stochastic process constraints that previous works could not satisfy. This is achieved by finding the right radius density function so that the corresponding elliptic distribution satisfies Kolmogorov's extension theorem. They then demonstrate how one may do Bayesian modeling and inference with this Q-EP process and provide experiments demonstrating their results.

**Strengths:**

* Very interesting stochastic process family that I can see having a lot of use cases
* Compelling experimental results showing the efficacy of their method

**Weaknesses:**

* The experimental results use pretty small images. How well do these methods scale up to larger images and high-dimensional domains more generally?

**Questions:**

* How should one choose q? Should q=1 always be the default starting point?

**Limitations:**

The authors have adequately addressed the limitations.

---

> ### Author Rebuttal · Authors · 2023-08-03
>
> We thank the reviewer for the nice comments and the generous support.
>
> We have three CT examples in Section 4.2: the Shepp–Logan phantom is of size $128\times 128$ and the other two human body parts CT images are of size $512\times 512$. They are pretty standard sizes for images in machine learning. Note, the discrete dimension $d=128^2$ or $512^2$. We need to specify covariance matrices of size $d\times d$ for GP and q-EP which are extremely large ($512^2\times512^2=68,719,476,736$). Therefore, we did dimension reduction by partial Eigen-decomposition (taking the first $L=2000$ largest eigenvalues) using randomized algorithms (Joel A. Tropp, ACM 204 CalTech). If the reviewer refers "larger images" to super-resolution satellite images, the methodology should still work in theory, but care needs to be taken when implemented in powerful GPUs or TPUs.
>
> As commented in the conclusion, $q=1$ is often adopted for q-EP (and Besov process as well) to impose more regularization as opposed to $q=2$ which corresponds to GP. We studied the effect of the parameter $q$ in Figure C.6 in the supplementary materials by varying $q\in (0,2]$. We found that smaller $q$ led to sharper reconstruction of the blurred image of satellite. In general, we could do cross validation to choose $q$ or even impose a hyper-prior on $q$ for a fully Bayesian treatment, which is expected to be more complicated and will be a good future direction.

---

> > ### Comment · Reviewer_n1kr · 2023-08-19
> > **Reply to authors**
> >
> > Thank you for your response. I will maintain my score.

---

### Official Review · Reviewer_Krqg · 2023-07-06

**Soundness:** 3 good
**Presentation:** 3 good
**Contribution:** 3 good
**Rating:** 5
**Confidence:** 3

**Summary:**

Motivated by the correspondence between Gaussian Process priors and ridge regularization for non-parametric regression problems, the authors in this paper develop a stochastic process prior, namely the $\textbf{Q-exponential (Q-EP) process}$, which can correspond to $\ell_q$-regularization. Specifically, by starting from multivariate  $q$-exponential distributions, the authors verify a Kolmogorov Consistency criterion to eventually develop the aforementioned process. Subsequently, the authors further provide justifications how the (Q-EP) process process allows a more flexible modeling of covariance kernels compared to parallel Besov processes also designed for similar tasks and derive a posterior predictive formula. Lastly, the benefits of the prior are demonstrated through numerical examples pertaining to problems in functional data analysis, image reconstruction, and solving inverse problems.

**Strengths:**

Offers a detailed probabilistic description and construction of a stochastic process prior for non-parametric Bayesian regression which (i) corresponds to $\ell_q$-regularization; (ii) is developed from easy to understand first principles; and (iii) allows easy computation of posterior predictive distributions.

**Weaknesses:**

Theoretically, it was not discussed in which type of non-parametric problems and function spaces one might get better rates of posterior contraction using the developed process type priors compared to Gaussian process priors. Most of the theory is basic probabilistic calculations and quite stratighforward.

**Questions:**

There is substantive literature on "minimax optimal" posterior contraction over standard function spaces (e.g. Sobolev, Besov etc.) results for suitable designed Gaussian process priors  -- can the authors discuss whether there are some set ups of non-parametric regression problems where some version of the priors developed here offers optimal posterior contraction over some function spaces?

**Limitations:**

None noted.

---

> ### Author Rebuttal · Authors · 2023-08-03
>
> Thanks for raising good points in the "Weaknesses" and "Questions" sections. Q-EP is proposed as a nonparametric prior for flexible Bayesian models including regression, classification, density estimation, inverse problems, etc. The motivation is to impose more regularization than Gaussian process (GP) on function spaces while enjoying the similar tractability (on correlation and prediction) as GP. The function space is assumed to be $L^q$ and we require the kernel function $\mathcal{C}$ is a trace-class (having summable eigenvalues) so the quadratic $r(u)=\langle u, \mathcal{C}^{-1}u\rangle$ and hence the process is well-defined.
>
> We are aware of Professor Ghosal, Professor Van Der Vaart and their collaborators' seminal works on posterior contraction theory on non-parametric Bayesian models. We actually have been working on similar theories for q-EP and a spatiotemporal Besov extension that relies on q-EP. However, given the limited space of 9 pages, we decided to focus on the introduction of q-EP, its application to nonparametric modeling and demonstration of its advantages over GP when modeling subjects with abrupt changes or sharp contrast (such as edges in image). We leave the rigorous treatment of contraction theory to a journal submission.
>
> We thank the reviewer for raising such a good question on comparing the contraction rates between q-QP and GP and we will investigate it in our ongoing work.

---

### Official Review · Reviewer_av1p · 2023-07-21

**Soundness:** 3 good
**Presentation:** 2 fair
**Contribution:** 3 good
**Rating:** 7
**Confidence:** 4

**Summary:**

As a generalization of GP, it is important to construct a stochastic process (prior) that can express various degrees of smoothness. As such process, the Besov processes have been proposed but they are defined in the form of a series expansion, and the corresponding probability distribution is not given in an easy-to-handle form.
In this work, the q-exponential process is proposed, which addresses the drawback of the conventional Besov process. A probability distribution called the q-exponential distribution (q-ED) that facilitates sampling from the Besov process is proposed. It is extended to multivariate in a consistent manner in the sense of marginalization. q-ED is expressed as an elliptic distribution, which also satisfies the exchange rate, thus satisfying the conditions of Kolmogorov's extension theorem, and the existence of the corresponding stochastic process is guaranteed. The KL expansion shows that the q-exponential process (q-EP) corresponding to the q-ED can be defined and has a series representation almost equivalent to the Besov process.
The q-EP has the strong advantage that the posterior predictive distribution can be constructed by MCMC, and Bayesian regression can be performed.

**Strengths:**

It is an important contribution from the viewpoint of statistical modeling to define an explicit probability distribution as q-ED and to provide a sampling method for Besov processes described in series representation. q-ED is firstly defined as a one-dimensional probability distribution and then is extended to a multivariate distribution paying attention to consistency. The method of extending to a stochastic distribution and the method of performing MCMC inference (posterior distribution calculation) based on stochastic representation is standard and reasonable in the context of SDE simulation (e.g., a scaled mixture of normals).
The proposed q-EP will have wide applicability and some successful application results are presented in the paper.

**Weaknesses:**

Minor issue:
It is claimed that "we have less control on the correlation strength once the orthonormal basis is chosen". It is true but when we are free to choose the orthonormal basis, I feel the conventional Besov process has equivalent or even better freedom for specifying the correlation structure to the proposed q-EP. I acknowledge that specifying the correlation structure via the covariance function is handy, but in terms of the degree of freedom, it is fair to consider the case we can freely choose the basis.

It is claimed in line 218 that "codes will be publicly available at to-be-released", hence currently I cannot evaluate the reproducibility of the experimental results.

Very minor presentation issues:
- Eq.(5) and elsewhere, p(u) should be treated as the function of u, not r. I mean, for example in Eq.(5),
p(u) = k_d |C|^{-1/2} g(r(u)).

- F_i in Theorem 3.1. should be defined.
- \mathbb{R}^k in Eq.(7) should be \mathbb{R}^{n}

- References should be properly described. For example, ref [1] does not have a link to the project, and ref [12] lacks bibliographic information.

**Questions:**

Q1: In my understanding, the proposed q-EP is essentially equivalent to the Besov process, and the major difference is its explicit control of the covariance structure. If this is the case, what is the main reason for the performance gap between Besov and q-EP in experiments, particularly for image reconstruction? If not, please reveal the difference between Besov proc. and q-EP. In particular, does q-EP includes Besov proc as a special case?

Q2: Related to the problem mentioned in 6.Weakness, I'm not sure the experimental setting in subsection 4.1 is fair. For the Besov process, the Fourier basis is chosen and fixed. It is plausible but the inferior performance of the Besov process could be contributed to the bad choice of the basis. Please validate this experimental setting in more detail.

Q3: I don't understand the meaning of the second sentence in "Introduction". How the "High-dimensional objects" can be viewed as "evaluation of proper functions", and what is the "proper function"?

Q4: What is the difference between "d^{\star}" in line 34 and "d" in other places?

**Limitations:**

I'm concerned that the scale mixture expression is not possible for all q.
See, e.g.,
M. West, ``On scale mixtures of normal distributions'', Biometrika (1987).
I found that the range 0 < q < 2 is specified in the Appendix, but should be explicitly stated in the body of the paper to make the applicability of the methodology clear.

---

> ### Author Rebuttal · Authors · 2023-08-03
>
> We thank the reviewer for supporting the contribution and the potential impact of q-EP. We agree with the reviewer that the correlation strengthen of Besov process can be configured through the choice of basis functions $\{\phi_\ell(x)\}$, as spelled out in Equation (12). But it is just less straightforward than specifying in the kernel function $C(\cdot,\cdot)$ as usually done in Gaussian process. We will reword the relevant sentences to reflect this.
>
> The Github repository hosting all the codes is currently private but will be made public when the work gets published. However, along with the submission, we provided a zip file in the supplementary materials containing Python codes for demo and an example of reconstructing blurred image of satellite. All the codes in the supplement are anonymous and accompanied with a `readme` help document so the reviewer should be able to reproduce some results by following the instruction. We hope the reviewer can understand and find it helpful.
>
> We thank the reviewer for careful reading and we will fix all the minor issues raised in "Weaknesses". We also appreciate all the questions and now answer them one by one:
>
> Q1) Yes, the statement about the relationship between q-EP and Besov process is right. As highlighted in the introduction, q-EP can be viewed a probabilistic definition of Besov process with explicit specification of correlation strength and tractable prediction formula. Their representation equivalence is further elaborated in Theorem 3.4 and Remark 2. Due to their difference in the mathematical format (probabilistic distribution vs series representation), they behave differently in numerics. We found q-EP is superior than Besov in relatively lower-dimensional cases (See Figures 3,4 and Tables 1,2) and they become more similar when the dimensions go much higher and dimension reduction (partial eigendecomposition of C) is implemented (See Figure5 and Table C.2). We agree that it is an interesting phenomenon and will investigate it in future work.
>
> Q2) We tried multiple wavelet bases including Hard, Shannon, Meyer and Mexican Hat etc. but none of them generated better result. We could include these reconstruction results in supplementary materials in the revision.
>
> Q3) Sorry for the confusion. What we tried to say is that each image can be viewed as a function defined on a bounded domain whose values are the pixels. We will revise this sentence to avoid confusion.
>
> Q4) $d^\star$ in line 34 is the dimension of space where subjects of interest are defined. For example, in our numerical examples, $d^\star=1$ for the time series and stocks and $d^\star=2$ for various CT images. While $d$ in other places refers to discrete dimension of the processes. For example, $d=N=200$ is the number of time points for the time series and $d=n^2=128\times 128$ is the image size for CT images. We will clarify them in the revision.
>
> Also thanks for pointing out the condition $0<q<2$ for the scale mixture result only mentioned in the appendix. We will make it explicit in the main text when revising the paper.

---

> > ### Comment · Reviewer_av1p · 2023-08-12
> >
> > I appreciate the authors' response.
> > My concerns including the reproducibility issues and equivalence to the Besov process in series representation are resolved; I'm now more confidently support this work. I raised score from 6 to 7.

---

> > > ### Author Response · Authors · 2023-08-13
> > >
> > > We are very thankful for the reviewer's support! Really appreciate all the constructive comments.

---

### Official Review · Reviewer_Yr66 · 2023-07-27

**Soundness:** 3 good
**Presentation:** 2 fair
**Contribution:** 3 good
**Rating:** 5
**Confidence:** 2

**Summary:**

This paper proposed a new random process prior that corresponds to estimating parameters with $\ell_q$ penalty. The process, named Q-EP, can be used to provide a shaper penalty than the standard Gaussian process. Empirical experiments show the practical use case for Q-EP.

**Strengths:**

The derivation of Q-EP looks solid to me. I am not sure about the novelty of the work.

The experimental results look convincing, though I do not know if there is a standard benchmark.

**Weaknesses:**

It is hard for people unfamiliar with the field to understand the paper. In particular, the authors do not give a fair amount of text to explain the background. It is hard to relate the abstract math in the introduction with the examples provided (e.g. Figure 1). There is no clear statement claiming the connection between Q-EP with the $L_p$ regularization. The Bayesian model is not introduced until page 6. In sum, I suggest a major rewriting of the paper, potentially with a different organization.

**Questions:**

What is your insight when comparing the Bayesian models with various denoising works in deep learning? I believe they are very different in many aspects.

**Limitations:**

I do not see any potential negative societal impact of this paper.

---

> ### Author Rebuttal · Authors · 2023-08-03
>
> We appreciate the reviewer's critics. As mentioned in the introduction, the novelty lies in the first probabilistic definition of Besov process (which is widely used in imaging analysis and Bayesian inverse problems) with explicit specification of correlations and tractable prediction formula.
>
> For nonparametric regression with novel priors, mean squared error (MSE) (or root MSE, RMSE) and log-likelihood (LL) (See Table 1) are standard measures to compare, as included in the paper by Professor Zoubin Ghahramani's group [37], the paper by Bankestad et~al [4], and the seminar book by Rasmussen and Williams [Gaussian Processes for Machine Learning, 2006]. We also added multiple standard quality metrics in imaging analysis (PSNR,SSIM, HaarPSI as in Table 2). They all support our claimed numerical advantages for q-EP. While we are open to any other metrics the reviewer might suggest, we are also willing to defend against "limited evaluation", given 4 time series examples, 3 CT image reconstructions and one Bayesian inverse problem regarding an advection-diffusion equation reported in this paper.
>
> Regarding the connection between q-EP and $L_q$ regularization, we kind of assumed for granted and that might have caused some confusion. It is similar to the relationship between GP and $L_2$ regularization: negative density of Gaussian distribution yields the $L_2$ regularization term usually added to the objective function, that is, $\frac{1}{2}\Vert u-\mu\Vert^2$. In q-EP scenario, that $L_q$ regularization term is $\frac{1}{2} r(u)^{\frac{q}{2}}$ with $r(u)=\langle u-\mu, \mathcal{C}^{-1}(u-\mu)\rangle$ plus some other comparatively smaller term $\log r$. We will add explicit explanation and more background relating the math to Figure 1 to the main text.
>
> Regarding the structure, we admit that we spent some space on explaining the marginalization consistency -- we did that to emphasize its importance and  the univariate q-exponential distribution may fail to generalize to a valid stochastic process without care, as happened in other literatures -- we think that is part of our novelty. The Bayesian model is discussed after the new prior q-EP is fully introduced. This is in the same spirit of Rasmussen and Williams' GP book, which does not jump to the model at beginning.
>
> We thank the reviewer for this good question. First, the paper introduces a nonparametric modeling tool with the novel q-EP prior imposing more regularization than GP. It can be applied to imaging analysis but not limited to image denoising. Second, one of the advantages of the proposed Bayesian models over the majority of optimization based deep learning techniques is the uncertainty quantification (UQ) (refer to Figures C.3 and C.4). UQ is of scientific interest and is the natural byproduct of Bayesian approaches however not the main focus of many vanilla versions of denoising works in deep learning. Last but not the least, these two approaches can interact and evolve to new methods. For example, the q-EP prior can provide a new loss function other than MSE for training the denoising neural network (NN). On the other hand, there might exist certain format of NN whose limiting behavior mimics q-EP prior, similar to the relationship between DNN and GP [Neal 1994a, Lee et al 2018 ICLR]. This is also an interesting direction the authors would like to pursue.

---

> > ### Comment · Reviewer_Yr66 · 2023-08-21
> >
> > Thanks for the clarification. I have raised my score accordingly. Nevertheless, I still recommend a thorough modification of the paper so it can be accessible to readers unfamiliar with the topic. The organization of a paper is generally different from the organization of one chapter in a book.

---

> > > ### Author Response · Authors · 2023-08-21
> > >
> > > We thank the reviewer for raising the score. We are glad that the reviewer has accepted our clarification on more important questions such as connection between Q-EP with the $L_p$ regularization, sufficient numerical evaluations, etc.
> > >
> > > Regarding the structure, we agree with the reviewer on the statement "The organization of a paper is generally different from the organization of one chapter in a book." (We cited Rasmussen and Williams' GP book to explain our writing style but did not mean to literally follow its organization). To improve the paper's clarify, We will take the reviewer's suggestion to revise the introduction with more background and elaboration, as also mentioned by Reviewer V1qe. We would also appreciate if the reviewer could elaborate "a thorough modification" with some more specifics.

---

### Official Review · Reviewer_V1qe · 2023-07-27

**Soundness:** 3 good
**Presentation:** 3 good
**Contribution:** 3 good
**Rating:** 6
**Confidence:** 1

**Summary:**

The authors propose the 'q-exponential process', a stochastic process interpretation of Lq function regularization, which can induce sparsity in the solutions to optimization problems and can be used as a functional prior for Bayesian applications in time series regression, image reconstruction, and other applications.

**Strengths:**

The proposed q-exponential process has several important benefits:
- enforcing sparsity or more sharpness in function space
- control of correlation structure, similarly to GP
- flexibility in choice of kernel
- conjugacy for posterior prediction with appropriate likelihoods, in contrast to the Besov case

**Weaknesses:**

The paper could provide more background on the problem setting and Besov processes, as the current introduction is difficult to follow.

**Questions:**

It may be interesting to further unpack Remark 3 regarding the similar mean & covariance for Q-EP and GP, for example showing how this affects predictions for Q-EP and GP in a tractable toy example.

**Limitations:**

The authors note the need for grid search over hyperparameters such as regularity parameter and smoothness parameter.    The method is tested only for q=1 and it would be interesting to explore other values.

---

> ### Author Rebuttal · Authors · 2023-08-02
>
> We thank the reviewer for the suggestion on more background about Besov processes. In addition to the existing introduction which includes its mathematical definition, we will elaborate more on its implication and applications on imaging analysis.
>
> We also appreciate the reviewer's advice on expanding Remark 3. In the revision, we will restore more mathematical details that were in the initial draft but hidden from the submission for brevity. However, we have two tractable toy examples in section 4.1 for which we compare the predictions for Q-EP and GP in Figure 3(b) and in Figure C.2(b) as well.
>
> Regarding the limitation, we emphasized in the manuscript that $q=1$ is the case often opted in (Conclusion). We did explore the effect of regularization parameter $q$ for a spectrum of different values in Figure C.6 and mentioned that briefly in the paragraph "Connection to existing work" in the introduction. We will make it more explicit with proper emphasis.

---

### Author Rebuttal · Authors · 2023-08-03

We thank all the anonymous reviewers for their careful reading, constructive advices and critics. All the reviews acknowledge the novelty of the proposed q-EP as a nonparametric prior and its potential impact in statistics and machine learning applications. Some demand clarification and presentation improvement, and others suggest theoretic exploration. All their comments and opinions are highly appreciated. Below we have made one-to-one responses and hope the reviewers and program/area chairs can find them useful when evaluating the paper. We will be happy to discuss more with reviewers in the discussion stage. Thanks!

---

### Decision · Program_Chairs · 2023-09-21

**Decision:**

Accept (poster)

**Comment:**

Strengths:
- The paper presents a novel Q-exponential process that brings several advantages to nonparametric statistical modeling.
- The methodological derivation is solid, and the application of the method appears to be successful in particular contexts.

Weaknesses:
- Multiple reviewers mention that the paper is not easily accessible due to insufficient background explanation, particularly about the problem setting and Besov processes.
- Concerns are raised about the theoretical depth of the paper, especially in comparison to Gaussian process priors.
- Reviewers ask for further clarity on the relationship between Q-EP and Besov processes, implications when comparing with Bayesian and deep learning models, and the settings where the proposed method might be advantageous. These areas need to be addressed.

Overall, the paper is expected to be a strong and impactful contribution to the field, and recommended for acceptance.  Howerver, the authors should address the raised concerns in the final version.